# Source attribution of European surface $O_3$ using a tagged $O_3$ mechanism

Aurelia Lupaşcu[1] and Tim Butler[1,2]

[1]Institute for Advanced Sustainability Studies (IASS), Potsdam, 14467, Germany
[2]Freie Universität Berlin, Institut für Meteorologie, Berlin, Germany

**Correspondence:** A.Lupascu (Aurelia.Lupascu@iass-potsdam.de)

**Abstract.**

Tropospheric ozone ($O_3$) is an important air pollutant that affects human health, ecosystems, and climate. The contributions of $O_3$ precursor emissions from different geographical source regions to the $O_3$ concentration can help to quantify the effects of local versus remote transported precursors on the $O_3$ concentration in a certain area. This study presents a "tagging" approach within the WRF-Chem model that attributes $O_3$ concentration in several European receptor regions to nitrogen oxides ($NO_x$) emissions from within and outside of Europe during April-September 2010. We also examine the contribution of these different precursor sources to various $O_3$ metrics and their exceedance events. Firstly, we show that the spatial distributions of simulated monthly mean MDA8 from tagged $O_3$ sources regions and types for late spring, summer and early autumn 2010 varies with season. For summer conditions, $O_3$ production is dominated by national and intra-European sources, while in the late spring and early autumn intercontinental transported $O_3$ is an important contributor to the total $O_3$ concentration. We have also identified shipping activities in the Mediterranean Sea as an important source of $O_3$ for the Mediterranean countries, as well as the main contributor to high modelled MDA8 $O_3$ concentration in the Mediterranean Basin itself. Secondly, to have a better understanding of the origin of MDA8 $O_3$ exceedances, we compare modelled and observed values of MDA8 $O_3$ concentration in the "Po Valley" and "Germany-Benelux" receptor regions, revealing that the contribution from local sources is about 41 % and 38 % of modelled MDA8 $O_3$ during the exceedances days respectively. By examining the relative contributions of remote $NO_x$ sources to modelled and observed $O_3$ exceedance events, we determine that model underrepresentation of long-range $O_3$ transport could be contributing to a general underestimation of modelled $O_3$ exceedance events in the Germany-Benelux receptor region. Thirdly, we quantify the impact of local vs. non-local $NO_x$ precursors on $O_3$ production for each European receptor region using different $O_3$ metrics. The comparison between mean, MDA8 and $95^{th}$ percentile $O_3$ metrics accentuate the importance of large contributions from locally-emitted $NO_x$ precursors to the high-end of the $O_3$ distribution. When we compare the vegetation and health metrics, we notice that the SOMO35 and AOT40 indexes exhibit rather similar behaviour, while the W126 index accentuates the importance of local emissions. Overall, this study highlights the importance of a tagging approach to quantify the contribution of local and remote sources to the MDA8 $O_3$ concentration during several periods as well to different $O_3$ metrics. Moreover, this method could be applied to assess different mitigation options.

# 1 Introduction

Tropospheric ozone ($O_3$) is formed primarily during the oxidation of volatile organic compounds (VOC) in the presence of nitrogen oxides ($NO_x$) and sunlight. Ground-level $O_3$ is an important air pollutant that damages human health (Fleming et al., 2018) and vegetation (Mills et al., 2018). It also affects the radiative forcing (e.g. Ramaswamy et al., 2001; Stevenson et al., 2013), and therefore contributes to climate change. Impacts of $O_3$ on human health are associated with lung disease, chronic disease and death from respiratory ailments. To protect human populations from exposure to high levels of $O_3$, the World Health Organization (WHO, 2006, 2017) recommended an air quality guideline for ozone in which the maximum daily average 8-h (MDA8) for $O_3$ should not exceed 100 $\mu g\ m^{-3}$. The European Environmental Agency (EEA, 2017a) reported that the EU long-term objective target concentration of 120 $\mu g\ m^{-3}$ is often exceeded and that more than 90 % of the urban population of the European Union was exposed to $O_3$ levels higher than the stricter recommendation set by the WHO. A 2010 report from HTAP (HTAP, 2010) shows that the observed baseline $O_3$ concentrations (concentrations without the contribution from local anthropogenic emissions) have increased throughout the last several decades since overall global anthropogenic emissions of $O_3$ precursors have increased. However, a more recent study by Gaudel et al. (2018) has established that the global surface $O_3$ trends exhibit high variability, and depend on several factors such as season, region, elevation and proximity to fresh ozone precursor emissions. However, since the network capable of monitoring ozone levels is sparse, it is difficult to quantify the $O_3$ changes on a global scale. Satellite-derived $O_3$ measurements can be used to quantify changing levels of $O_3$, but Gaudel et al. (2018) showed that these products are not capable of quantifying significant trends. Surface $O_3$ pollution due to urbanization and motorization processes are serious challenges for large cities (e.g. Chan and Yao, 2008; Folberth et al., 2015; Li et al., 2017, 2019). Paoletti et al. (2014) showed that in Europe and the United States of America, the average $O_3$ concentration in the cities has increased at a faster rate than those observed in rural areas. Fleming et al. (2018) showed that the 4th highest daily maximum 8-hour $O_3$ (4MDA8) is more ubiquitous at urban sites than at non-urban sites. This leads to a worsening of general air quality that, ultimately, affects human health and ecosystems (Paoletti et al., 2014; Monks et al., 2015; WHO, 2017; Fleming et al., 2018; Mills et al., 2018). To improve the air quality in certain areas, it is important to know the extent to which different precursors ($NO_x$ and VOCs) contribute to tropospheric $O_3$ concentrations.

Information regarding levels of $NO_x$ and VOC emissions and weather conditions enhances our ability to predict the formation of tropospheric $O_3$. The continuous development of chemical transport models can lead to a better understanding of the processes that contribute to high-$O_3$ episodes. Knowing the impacts of $NO_x$ and VOC emissions from sources such as surface anthropogenic activities, fires, soil, and the stratosphere on total $O_3$ production can help authorities develop strategies aimed at reducing the impact of high levels of $O_3$ on the well-being of both humans and ecosystems. Several approaches have been used to determine the extent to which individual sources contribute to total levels of $O_3$. For example, perturbation of different emission categories have allowed scientists to make estimations regarding the contributions of individual sources of $O_3$ to total $O_3$ levels (e.g. Fiore et al., 2009).

Tagging techniques have also been used in modelling studies to determine source/receptor relationships and how individual sources of pollutants contribute to total pollution levels at given locations. Pollutants with relatively low chemical reactivities,

such as carbon monoxide (CO), can be "tagged" according to their emission sectors or regions for attribution studies (e.g. Pfister et al., 2011). Sudo and Akimoto (2007), and Derwent et al. (2015) used $O_3$ tracers tagged by their region of formation to show that intercontinental transport of $O_3$ can occurring from polluted source regions, such as North America and East-Asia, appears to be the most important source of tropospheric $O_3$ in Europe. Other studies, including those of Wang et al. (2009) and Grewe

et al. (2010, 2012, 2017) have used tagging methods to identify the contribution of individual sources of $O_3$ to overall levels. This method is especially useful since it can track emitted $NO_x$ species during transport and chemical processing. Moreover, Grewe et al. (2012) showed the impact of the tagging method on mitigation measures, while Dahlmann et al. (2011) examined the contribution of $O_3$ sources to $O_3$ radiative forcing. Work by Emmons et al. (2012) and Butler et al. (2018) describe a procedure for tagging $O_3$ produced from $NO_x$ sources through updates to the MOZART chemical mechanism, and Butler et al.

(2018) expanded the tagging technique to account for VOC sources.

Based on the work of Emmons et al. (2012), Pfister et al. (2013) and Safieddine et al. (2014) were able to use the WRF-Chem regional model to quantify the contribution of inflow (tagged $O_3$ and odd nitrogen species entering into the regional domain at the lateral boundaries) and of anthropogenic $NO_x$ precursors (named $NO_x$ in the following) on the surface $O_3$ levels. Using a slightly different methodology, Gao et al. (2016) have implemented within WRF-Chem framework a tagging method based

on Ozone Source Apportionment Technology (OSAT) (Yarwood et al., 1996) incorporated in the Comprehensive Air quality Model with extensions (CAMx).

Much effort has been focused understanding the origin of tropospheric $O_3$ and the key role played by the intercontinental transport, the contribution of stratospheric $O_3$ intrusion, and of different emissions sources to tropospheric $O_3$ concentration in a wide range of receptor regions. To better understand the changes in air pollution levels, it is necessary to know the relationship

between levels of an emitted species and its atmospheric concentration. When this information is available, it is possible to quantify the contribution of different emission precursor sources to overall $O_3$ concentration levels at a particular receptor location. For this purpose, we followed a strategy outlined in Emmons et al. (2012) and Butler et al. (2018) to implement a tagging technique into the regional WRF-Chem model. The model can be used to quantify source contributions to the tropospheric $O_3$ concentration, by "tagging" $NO_x$ emissions, and corresponding products so that they can be traced to the final

production of $O_3$.

When studying the effects of $O_3$, the impact of the compound on humans and vegetation is of the utmost importance. Therefore, several exposure indexes have been defined to describe the relationship between $O_3$ and both human health and agricultural crop yield that are based on hourly averaged data. Musselman et al. (2006), Agathokleous et al. (2018), and Lefohn et al. (2018) review literature describing $O_3$ metrics. Additionally, a work by Paoletti et al. (2007) has provided a list of

common $O_3$ exposure metrics used to assess risk to human health and vegetation. Here we use some well-known $O_3$ metrics, such as MDA8, SOMO35, AOT40, and W126. The MDA8 index has been defined as the maximum daily average 8-h (MDA8) $O_3$ values (ppb) (Lefohn et al., 2018). SOMO35 (WHO, 2001) has been determined by European protocols (EU directive 2008/50/EC, 2008) and is defined as the annual sum of MDA8 $O_3$ with a cut-off of values of 35 ppb. Both MDA8 and SOMO34 are health-related metrics. The AOT40 and W126 vegetation metrics have been used to regulate air pollution in both Europe

(EU directive 2008/50/EC, 2008) and the United States (U.S. EPA regulations https://www.gpo.gov/fdsys/pkg/FR-2015-10-

26/pdf/2015-26594.pdf). European legislation (EU directive 2008/50/EC, 2008) AOT40 metric is measured throughout daytime periods from May to July (growth season) and has a defined target limit of 18000 $\mu$g m$^{-3}$ h (9000 ppb $-$ hours) and a long term objective of 6000 $\mu$g m$^{-3}$ h (3000 ppb $-$ hours). A standard of 15 ppm $-$ hours has been defined for the seasonal W126 index, which is averaged over three years. These metrics have been used to assess the impact of mitigation strategies (Avnery

et al., 2013), the impact of industry on air quality management issues (Vijayaraghavan et al., 2016), and the impact of high O$_3$ levels and temperatures on crops (Tai and Val Martin, 2017).

In this paper, we use a tagged O$_3$ mechanism in the WRF-Chem model to understand the contribution of emitted O$_3$ precursors from different geographical source regions and types on the modelled O$_3$ concentration in several European receptor regions. In Section 2 we discuss the details of implementing this tagging technique, and describe changes made to both the

chemical mechanism and WRF-Chem code. Section 2 also describes the WRF-Chem configuration, simulation design, and input data used in the study. An analysis of the WRF-Chem simulation is presented in Section 3, while Section 4 summarizes our findings.

## 2 Model simulation

### 2.1 Tagging technique

To perform a WRF-Chem model simulation using a tagging approach, several changes must be implemented in the model code to accommodate additional tracers and reactions representing tagged constituents. Butler et al. (2018) describes in detail how the tagging technique was implemented in the Community Earth System Model. The tagging technique used in this study is based on the same approach and uses the same modified version of the MOZART chemical mechanism. Further detail on how the chemical mechanism was extended can be found in Butler et al. (2018).

To use the NO$_x$ tagging mechanism, a new chemistry option was added in the namelist.input file: chem$_-$opt=113 and through the code. The coupling of the new chemical scheme with microphysics and radiative processes requires several modifications to the code: 1) The first step is to create a new chemistry option. The package mozart$_-$tag$_-$kpp (chemopt==113) has been added to ~/WRFV3/Registry/registry.chem together with new model variables for tagged NO$_x$ species (e.g. O3$_-$X$_-$INI, O3$_-$X$_-$STR, etc). For this purpose, the pre-processing software described in Butler et al. (2018) was adapted to produce

a new chemical mechanism; 2) The new chemistry package is a KPP option. Therefore, we created a new subdirectory in ~/WRFV3/chem/KPP/mechanisms/ directory containing the files (*.spc, *.eqn, *.kpp, and *.def) which defined the chemical model species and constants, chemical reactions in KPP format, model description, computer language, precision, and integrator.

The new chemistry option considers a large number of species and reactions; therefore we exceeded hard-coded limits

that the KPP chemical preprocessor, version 2.1 (Sandu and Sander, 2006) allows. To overcome these limits, we increased MAX$_-$EQN and MAX$_-$SPECIES in the header file gdata.h, located in ~/WRFV3/chem/KPP/kpp/kpp-2.1/src. Further, we updated the subroutines in the ~/WRFV3/chem directory to consider the new chemistry package. The modules that we modified are described in the Appendix.

Although WRF-Chem uses the Advanced Research WRF (ARW) dynamic core in this simulation which conserves mass and scalar mass (Grell et al., 2005), the tagged $O_3$ species are advected independently. Thus, numerical errors associated with the advection scheme led to gradients in the sum of tagged species concentration compared to the "real" concentration; therefore, the relationship between these variables is not conserved. Since the advection scheme fails to reproduce the expected solution (in which the sum of the tagged species concentration at each grid point must be equal to "real" concentration), we solve this by fixing all undershoots and/or overshoots assuming that the sum of tagged species mass is proportional to the "real" concentration. This technique was also applied in Flemming et al. (2015), and Gromov et al. (2010).

Compared to Pfister et al. (2013) and Safieddine et al. (2014) work, the expanded tagging technique used in this study has the advantage that multiple tags can be defined in each model run.

## 2.2 Experimental setup

WRF-Chem version 3.7.1 was used for this study to account for the impact of different global and European $O_3$ precursor source regions to several European receptor regions during the April-September 2010 period. A single domain, that covers the area between $32°$ N and $70°$ N, and $29°$ W and $57°$ E, was used with 50-km grid spacing and 35 vertically-stretched layers from the ground up to 50 hPa. The physics options used for this study include the Morrison double-moment microphysics scheme (Morrison et al., 2009), the Grell-Freitas cumulus parameterization (Grell and Freitas, 2014), the Rapid Radiative Transfer Model (Iacono et al., 2008) for longwave and Goddard shortwave scheme (Chou and Suarez, 1994), the Yonsei University boundary-layer parameterization (Hong et al., 2006), and the Monin-Obukhov scheme for the surface layer (Jiménez et al., 2012). The initial and boundary conditions for meteorological fields are taken from the European Centre for Medium-Range Weather Forecasts (ECMWF) reanalysis. Anthropogenic emissions were obtained from the TNO-MACC III emission inventory for Europe (Kuenen et al., 2014). Because the model domain extends beyond the edges of the TNO-MACC III inventory, we used for completion emissions from the HTAP V2 inventory (http://edgar.jrc.ec.europa.eu/htap_v2). Biogenic emissions were computed on-line using the Model of Emissions of Gases and Aerosols from Nature (MEGAN) model (Guenther et al., 2006). The biomass burning emissions are based on Fire INventory from NCAR (FINN) (Wiedinmyer et al., 2011).

For this WRF-Chem simulation, the tagged MOZART chemical mechanism for $NO_x$ emissions (Butler et al., 2018) is used to represent the gas-phase chemistry. The photolysis rates were computed using the Fast Tropospheric Ultraviolet and Visible (FTUV) Radiation Model (Tie et al., 2003; Li et al., 2005). The dry deposition was calculated following the Wesely (1989) resistance method, while the wet removal scheme for the tagged MOZART chemistry is based on Neu and Prather (2012).

NOx emitted by several source regions and types are tagged and explicitly tracked using additional tagged reactions and tracers. Thus, we follow the contribution to the total ozone concentration from each specific emission source and type, from both within and outside the European model domain. Table 1 summarizes tagged sourceis regions and types that are used in this study. Using a division of source regions within the European model domain, 15 geographical source regions are specified in Table 1 and depicted in Figure 1. A similar division of European regions has been used by Christensen and Christensen (2007) and Otero et al. (2018) to address the main sources of uncertainty in regional climate simulations, as well as during the AQMEII project (i.e. Struzewska et al., 2015). Except for ALP, the source regions within the European domain are identical to

receptor regions. Given the complex topography of the ALP source region, we split this region into two receptor regions: the Po Valley region and the high Alps (regions above 1500 m elevation).

To represent the impact of transported $O_3$ from different regions outside of the domain, we used chemical boundary conditions derived from the extended CAM-Chem version 1.2 global simulations. Butler et. al (in preparation) used the tagging approach within the CAM-Chem model for several HTAP2 source regions such as: ASI (Asia), NAF (North-Africa), NAM (North-America), OCN (Oceanic sources), RBU (Russia, Belarus, Ukraine), and RST (rest of the world), as well as for several other source types: BIO (biogenic emissions), BMB (biomass burning emissions), LGT (lightning), and STR (stratospheric $O_3$). No overlap of source regions or types is allowed.

The BIO, BMB, LGT, and STR source types are also included in the tagged chemical mechanism used in this simulation, but without including them into the division of source regions; we refer to these sources as "other global source types" from here on. Ozone due to these other global source types can originate both from long-range transport from remote source regions through the lateral model boundaries as well as from precursor emissions within the European model domain.

For each receptor region, we analyse the impact of the $NO_x$ emissions coming from different source regions and types to the total $O_3$ concentration.

## 2.3 Ozone metrics

Using different metrics to assess the impact of $O_3$, we can determine which precursor sources most highly influence the accumulation of $O_3$ in different receptor regions, and thus to provide insights into which type of mitigation measures will be useful for a particular geographic area. These metrics include the mean $O_3$ concentration, the mean of MDA8, the cumulative exposure to mixing ratios above 35 ppb (SOMO35) (Colette et al., 2012), and the $95^{th}$ percentile for surface $O_3$. Neither the impact of $O_3$ exposure on trees, plants and ecosystems (W126) (Lapina et al., 2014), nor the AOT40 accumulation metric (the threshold is 40 ppb) were used to assess risk to vegetation from $O_3$ exposure (UNECE, 2010).

The European Air Quality Directive (EU directive 2008/50/EC, 2008) specifies that $O_3$ exposure should remain below a target MDA8 $O_3$ value of 120 $\mu$g m$^{-3}$, which can be exceeded up to 25 days per calendar year averaged over three years. The modelled daytime AOT40 (during local daylight hours 8 AM – 7 PM) was calculated according to Equation (1).

$$AOT40 = \sum_{i=1}^{90\,\text{days}} \left( \sum_{h=8}^{19} max\left(O_{3i,h} - 40,0\right) \right) \tag{1}$$

According to European legislation (EU directive 2008/50/EC, 2008), the AOT40 metric is accumulated over the daytime period from May to July (growth season) and it has a defined target limit of 18000 $\mu$g m$^{-3}$ h (9000 ppb − hours) and a long term objective of 6000 $\mu$g m$^{-3}$ h (3000 ppb − hours). W126, however, is described according to U.S. EPA regulations (https://www.gpo.gov/fdsys/pkg/FR-2015-10-26/pdf/2015-26594.pdf). A standard of 15 ppm − hours is defined for the seasonal W126 index, which is an average over a three-year period. For this study, the hourly surface $O_3$ tagged outputs for April

through September are used to calculate the highest 3-month W126 index values (see Eq. 2):

$$W126 = \sum_{i=1}^{90 \text{ days}} \left( \sum_{h=8}^{19} O_{3i,h} \cdot \left( \frac{1}{1 + (4403 \cdot e^{-126 \cdot O_{3i,h}})} \right) \right) \tag{2}$$

According to Lefohn et al. (1988), the W126 index includes all hourly $O_3$ values within the specified time range, although a lower weight is given to hourly $O_3$ concentrations below the inflection point of 65 ppb, while values above 90 ppb are weighted with a factor of almost one. SOMO35 (WHO, 2001) is defined as the sum of the MDA8 $O_3$ with a cut-off of 35 ppb (see Eq. 3). For this metric, the EU air quality directives do not prescribe a limit or a target values.

$$SOMO35 = \sum_{h=1}^{6 \text{ months}} max\,(\text{MDA8}_i - 35, 0.0) \tag{3}$$

The contribution of tagged $O_3$ is based on formulations of each metric and is calculated from the model output. In the case of the MDA8 and $95^{th}$ percentile metrics, we searched for the specific period in which calculated values of total $O_3$ concentration meet the requirements for the formulation of these metrics. Once this is identified, tagged $O_3$ concentrations are extracted for the same period which can then be used for further analysis. However, the contribution of concentration of tagged $O_3$ on cumulative metrics is slightly different, a large proportion of each tagged species is used to determine total $O_3$, as illustrated below for AOT40 at a specific time period:

$$AOT40_{tag} = \sum_{i=1}^{90 \text{ days}} max\left( (O_3 - 40.) \cdot \frac{O_{3,tag}}{O_3}, 0.0 \right) \tag{4}$$

Based on their formulation, we grouped metrics into either non-cumulative (mean $O_3$, MDA8, and the $95^{th}$ percentile) or cumulative (SOMO35, W126, and AOT40) categories. Since the latter metrics have different formulations (including hourly $O_3$ values above a specific threshold) and do not cover the same periods, to facilitate a more direct comparison between findings from multiple $O_3$ metrics, an analysis of the relative contribution of different source regions to total $O_3$ in each receptor region was performed. This was done using averaged values for non-cumulative metrics, and 6-month sums for SOMO35, AOT40, for cumulative metrics useful for evaluating effects on crops (cumulated over May-July period) and a maximum of 3-month sums for every consecutive 3-month period determined using the W126 index.

## 3 Results and discussions

Our discussion of the results of the model is focused on the April-September 2010 period. We first briefly evaluate the ability of WRF-Chem to reproduce meteorological parameters using measurements from the Global Weather Observation (GWO) dataset provided by the British Atmospheric Data Center (BADC), and observed $O_3$ concentrations using the measurements included in AirBase, a European air quality database (EEA, 2017b). We then provided a more detailed analysis of the contribution of different source regions and types to MDA8 values describing total $O_3$ for the analysed period.

## 3.1 Evaluation of meteorology and chemistry

Since the accurate simulation of meteorological parameters represents a key factor affecting the concentrations of trace gases, we briefly compare the modelled mean sea level pressure (MSLP), 2 m temperature (T2M), 10 m wind speed (WS10M) and direction (WD10M) variables against GWO measurement. Predicted model variables were then evaluated against observations using statistical scores that include normalized mean bias (NMB), and the correlation factor between simulated and measured values (r).

An extensive evaluation of WRF-Chem using the MOZART chemical mechanism to predict long term meteorological data and $O_3$ levels has been presented previously (Mar et al., 2016). The main differences between the set-up used in this study and the model described by Mar et al. (2016) include differences between the versions of the model used (3.7.1 vs. 3.5.1, respectively), horizontal resolutions (50kmx50km vs. 45x45km, respectively), microphysics (Morrison vs. Lin, respectively) and cumulus schemes (Grell-Freitas vs. Grell 3-D, respectively), simulation years (2010 vs. 2007, respectively), anthropogenic emissions inventory (TNO-MACC III vs. TNO-MACC II, respectively), and chemical input and boundary conditions (extended CAM-Chem version 1.2 with MOZART-4 vs. MOZART-4/GEOS-5 simulations found at http://www.acom.ucar.edu/wrf-chem/mozart.shtm respectively).

Due to the coarse resolution of our domain, the air parcel dynamics associated with the complex topography of mountainous areas was not properly reproduced. Thus, we assessed the ability of the model to reproduce the meteorological variables using only those sites located below 1500 m above sea level. MSLP data were well reproduced over the entire period (NMB = 0 % and r = 0.98). The model predicted T2M values well (r = 0.91), however temperature was underestimated by 3 % (see Table 2). WS10M was also fairly well reproduced both in terms of spatial and temporal variability (NMB = 8 %, r = 0.63). Yet, WD10M data could not be predicted as well as other meteorological variables (NMB = 13 %, r = 0.47), behaviour could be related to the existence of unresolved topography features (Jimenez and Dudhia, 2012). However, the model performance is similar to Mar et al. (2016) and Tuccella et al. (2012).

We also compared modelled MDA8 $O_3$ concentrations with observations provided by the publicly-available AirBase dataset. The relatively coarse resolution of the domain may not be representative of changes in local emissions when the measurements are taken from urban areas; therefore, to aid in the analysis, we used only those stations characterized as rural. As can be seen in Table 3, evaluation of the model over entire period revealed that the model performs quite well with respect to the prediction of concentration and temporal evolution. Mar et al. (2016) reported a mean bias (MB) value of 15.85 $\mu$g m$^{-3}$ and an NMB of 17% for the June–August 2007 period when the MOZART mechanism was used to assess the chemical performances of the model. For the same time period, we obtained an MB value of -5.92 $\mu$g m$^{-3}$ and an NMB value of -6.3%. Tuccella et al. (2012) reported an annual MB of -1.4 $\mu$g m$^{-3}$ when the RADM2 chemical mechanism was used to simulate a period throughout 2007. Month-to-month analysis (Table 3) shows that the model reproduces the $O_3$ concentration well compared to the Mar et al. (2016) and Tuccella et al. (2012). Even though the performance of the model in terms of temporal variation is relatively good (r values fall between 0.58 and 0.71), it mostly underestimated concentrations of $O_3$, except in September, when the model overestimated concentrations (NMB = 4.6%). Errors of the model may be explained by a wide range of uncertainties related

to modelled physical and chemical processes such as grid resolution, vertical and horizontal transport, boundary layer mixing, emission inventory, chemistry and photolysis rates, dry deposition, wet scavenging, etc. It is also possible that uncertainties in measurements contribute to observed errors. Since the focus of this study is on the contribution of different sources of precursors to the total tropospheric $O_3$ concentration of a particular area, a more thorough analysis of the ability of the model to reproduce the observed meteorological variables is beyond the scope of this paper.

### 3.2 Contribution of tagged precursor sources to the MDA8 $O_3$ mixing ratios

Figure 2 shows the spatial distributions of simulated monthly mean MDA8 values from tagged $O_3$ source regions and other global source types throughout late spring in 2010. The receptor regions shown were mainly influenced by the overseas combination of NAM, ASI, OCN, and RST sources that combined contribute from 23 % in the Po Valley to up to 53.6 % in the UKI region (see Table S1). $O_3$ from RST (a 7.5 - 15 % contribution) is the main source from overseas. $O_3$ from shipping NOx emissions advected through the model boundaries combined with $O_3$ produced from shipping $NO_x$ emissions in the Atlantic Ocean mostly affects Atlantic coastal countries (up to a 16.1 % contribution in the UKI region), yet a small contribution of ~4-5 % was also observed within inland regions. Long-range transport of $O_3$ from Asia and North America contributes significantly to total observed $O_3$ in Europe, accounting for 9.6 % of the total observed $O_3$ in ITA and up to ~22 % in UKI and SCA. After intercontinental transport, $O_3$ produced within Europe is an important source of $O_3$ in receptor regions, followed by $O_3$ coming from other global source types (LGT, BIO, and BMB). In general, for the April-May 2010 period, the contribution from the local sources to the total MDA8 $O_3$ mixing ratio in receptor regions falls within a range from 8.5 % (SCA) to 21 % (RBU) (see Table S1). Emissions from local sources do not only affect local $O_3$ mixing ratios, but also impact $O_3$ levels of bordering countries due to strong horizontal pollution transport. In all receptor regions, local anthropogenic sources have a lower contribution to MDA8 $O_3$ mixing ratios than the sum of $O_3$ due to anthropogenic sources in other European source regions and long-range transport of ozone from intercontinental source regions. The contribution of intercontinental transport to the total MDA8 $O_3$ mixing ratio in Europe is consistent with previously reported results, i.e. Fiore et al. (2009) and Karamchandani et al. (2017), while this study allows us to identify which anthropogenic sources exert a strong influence on MDA8 $O_3$ predicted in different regions. Using observations, Danielsen (1968), Thouret et al. (2006) showed that the transport of $O_3$ from the stratosphere also contributes to tropospheric $O_3$. Here, stratospheric $O_3$ contributes up to 7 ppb (12.5 % in SCA) to the total MDA8 $O_3$ mixing ratio, which is a finding similar to that reported by Derwent et al. (2015). A similar tagged system for predicting $O_3$ levels, using the CAM-Chem model (Butler et al., 2018), has also shown that stratospheric $O_3$ significantly contributes to the total tropospheric $O_3$ mixing ratio. The MOZART chemical mechanism used in this study does not explicitly treat stratospheric chemistry; thus surface stratospheric $O_3$ could be attributed to the vertical and horizontal transport of stratospheric $O_3$ and stratospheric tagged precursors species concentrations coming from the CAM-Chem extended model that enters the domain through lateral boundaries.

During June-August 2010, Western Europe was mostly influenced by a high-pressure system centered over the Atlantic (see Fig. S1). In the upper troposphere, a ridge influenced the vertical atmospheric structure, especially over southern Europe. Therefore, these "usual summer conditions" favoured the intrusion of warm air coming from Africa and the Arabian peninsula

and led to a warm and dry climate characterized by subsidence, stability, clear skies and high-intensity solar radiation. Hence, the photochemical formation of $O_3$ was enhanced, and influenced the stronger contribution of local emissions to the total mixing ratio compared to the previous period examined. Figure 3 depicts the average MDA8 $O_3$ for June-August 2010. For most regions, we notice that levels of $O_3$ produced from local sources from June–August compared with April–May were

enhanced (Figure 2). Local sources can contribute to more than 20% of the mean MDA8 $O_3$ mixing ratio (from 14.6 % in SCA to 35.7 % in the Po Valley, see Table S1). This shows that local sources play a strong role in the formation of $O_3$ throughout the June-August period, as has been previously shown by Jiménez et al. (2006) and Querol et al. (2018). Compared with late spring, the relative contribution of overseas sources decreased in summer, varying from 10.9 % in the Po Valley receptor region to 44.8 % in the UKI region in the month of July (Figs. 2 and 3; Table S1). We noticed the spread of $O_3$ produced from

European anthropogenic precursors over bordering regions compared with late spring 2010 (Figs. 2 and 3). The increase in average temperature combined with stable atmospheric conditions lead to an enhancement of the biogenic NO emitted into the atmosphere, especially in South-Eastern and Eastern Europe; thus, the BIO global source type contributes up to ~9 ppb (13.2 % of MDA8 $O_3$) in the RBU receptor region (see Fig. 3). The vegetation fires that took place across Russia in July and August (Gilbert, 2010; Huijnen et al., 2012) as well as in Portugal and Spain (European Commission, 2011) lead to increases in the

contribution of $O_3$ coming from BMB of up to 29 ppb (16 %) in the RBU receptor region and up to 8.5 ppb (2.3 %) in the IBE receptor region. BMB emissions contribute domain-wide more than 3 % (Po Valley), with the greatest impacts modelled over RBU, IBA, SEE, SCA, and TCA. The enhanced photochemical activity during summer combined with the weakening of stratospheric-tropospheric exchange reduces the influence of stratospheric $O_3$ from a domain-wide mean MDA8 $O_3$ mixing ratio of 4.4 ppb in the spring to 1.3 ppb in the summer (Figs. 2 and 3).

The decrease in photochemical activity in September 2010 is reflected in decreases in total $O_3$ mixing ratios compared with the summer of the same year, as well as in a reduction associated with the local source contribution to the total $O_3$ mixing ratio (Fig. 4). Thus, only in IBE, TCA, FRA, Po Valley, the high Alps, and RBU regions was contribution of local sources to total MDA8 $O_3$ higher than 20 % (Table S1). On the other hand, we noticed an increase in $O_3$ coming from anthropogenic overseas sources and from lightning in autumn, stressing that seasonal variations exist within the outflow from other continents. There

also is variation in the lifetime of $O_3$ which is shortest during the summer as a result of enhanced photolytic activity.

Although we have seen that long-range transport plays a major role in total $O_3$ mixing ratios, the tagging technique helps to gain more insight into which region of the world dominates these mixing ratios in spring or autumn. In early fall, the Western European receptor regions exhibit a slight increase of 1.6 % in $O_3$ mixing ratios coming from North America compared with spring, while the contribution of $O_3$ mixing ratios coming from other overseas sources decreases. This could be linked to the

prevailing westerly wind and the synoptic conditions seen during the first period of September, when the Azores High extended far to the east and north (Fig S1). This phenomenon creates conditions that are conducive to the transatlantic transport of American pollution in the eastern direction. For example, in autumn periods within the RBU receptor region, North-American and oceanic sources account for up to 14.6 % in spring and 11.4 % in autumn of the MDA8 $O_3$ mixing ratios.

Apart from local and other global souce types, $NO_x$ emissions from shipping activities in the Atlantic Ocean combined

with the oceanic $O_3$ from boundary conditions are an important source of $O_3$ that explains up to 16 % in late spring, 21% in

summer and 12% in early autumn of the MDA8 $O_3$ mixing ratio in the UKI, IBE, FRA, GEN, CEE, and SCA regions. Butler et al. (2018) showed that $O_3$ from oceanic sources reaches a minimum level in the North Atlantic Ocean during the summer, yet this study shows that in the UKI, IBE, FRA, GEN, CEE, and SCA receptor regions the oceanic $O_3$ contribution peaks in the summer. This implies that the nearby shipping emissions have a greater impact on oceanic bordering countries rather

than oceanic $O_3$ from boundary conditions. Furthermore, the $NO_x$ emissions from shipping activities in the Mediterranean and Black Seas account for up to 14 % in late spring, 19 % in summer and 11 % in early autumn of the MDA8 $O_3$ mixing ratio predicted in the receptor regions situated along the shore of the Mediterranean Sea, such as IBE, ITA, SEE, and FRA.

Our model results has shown that the highest MDA8 $O_3$ mixing ratios are predicted to occur over the Mediterranean Basin. This is due to the presence of favorable conditions for $O_3$ formation including the presence of small deposition sinks and

intense photochemistry (Figs. 2–4). Several studies, such as Safieddine et al. (2014), Tagaris et al. (2017), Mertens et al. (2018), Querol et al. (2018) and the references therein, have used source attribution methods to establish the origin of tropospheric $O_3$ observed over the Mediterranean Basin. The tagging technique used here shows that the $O_3$ from shipping activities in the Mediterranean and Black Seas (MBS) explains, on average, 15 % in late spring, 20 % in summer and 12 % in early autumn of total MDA8 $O_3$ predicted to accumulate within the MBS receptor region. These findings are similar to those of Aksoyoglu

et al. (2016) that showed these emissions accounted for 10-20 % of the mean $O_3$ in the Mediterranean in the summer of 2006. Moreover, Tagaris et al. (2017) have shown that shipping emissions explain up to 30 % the MDA8 $O_3$ simulated for July 2006 over the Mediterranean Sea. This study has shown that the shipping activities likely accounted for up to 35 % of the MDA8 $O_3$ near the Strait of Gibraltar (see Figure 5) during the April-September 2010 period. Shipping emissions contribute most highly to total $O_3$ in the Western Basin of the Mediterranean Sea. Aside from shipping activities, the other European source

regions have a localized contribution to total MDA8 $O_3$ predicted in the Mediterranean Sea. Thus, ITA, ALP, GEN source regions contribute mostly to the central basin; IBE and FRA are main contributors in the western basin and SEE and TCA predominantly contribute to the eastern basin. Natural sources contribute on average up to 10% of MDA8 $O_3$ in the western basin, and up to approximately 25 % of MDA8 $O_3$ in the eastern basin. The long-range of $O_3$ transport contributes up to 45 % along the North African shore and it exhibits a zonal pattern, with low mixing ratios occurring in the North and high mixing

ratios occurring south of the Mediterranean Sea, a trend mostly due to $O_3$ mixing ratios from NAF and RST sources.

### 3.3 Tagged ozone precursor contributions to exceedances of MDA8 target values – case study

As previously mentioned, the European Air Quality Directive (EU directive 2008/50/EC, 2008) has defined a target value of 120 $\mu$g m$^{-3}$ for the MDA8 $O_3$ concentration, which can be exceeded up to 25 days per calendar year (over a three-year span). In the following, we refer to values that surpass 120 $\mu$g m$^{-3}$ as exceedances, and values below 120 $\mu$g m$^{-3}$ as non-

exceedances. Figure S2 shows the spatial distribution of the number of exceedances observed and calculated throughout the April-September 2010 period for the AirBase rural stations. The observed MDA8 $O_3$ exceeds the target limits locally in Po Valley, Austria, and Germany; in coastal areas of Portugal, Spain, France and Italy; and inland areas of Poland and Slovakia. However, the modelled exceedances do not exhibit the same spatial pattern or intensity as observed values. Our use of tags allows for the identification of main source contributors to exceedances of modelled MDA8 $O_3$. Given the high number of

stations that measure O$_3$, for simplicity, we will discuss the source contribution to the MDA8 O$_3$ exceedances only for the Po Valley, high Alps, and GEN receptor regions.

Figure 6 exhibits the contribution of each tagged source and type to modelled and to observed MDA8 O$_3$ values. Samples were, in all cases, taken at the location of the measurement stations, throughout the April-September 2010 period. Figure 6 shows the average conditions during the exceedance of the MDA8 O$_3$ target value, and also, at times, occurred when the target value was not exceeded. To perform the source attribution for the observed values, we have scaled these values proportionally by the relative concentrations of each tagged O$_3$ tracer in our model output.

The relative contribution of emissions from different source regions to modelled and to observed MDA8 O$_3$ values, after being scaled to account for the contribution of modelled sources of O$_3$ types is generally similar for Po Valley and GEN receptor regions (see Fig. 6). In the Po Valley, we can pinpoint the main remote contributor as being MBS (see Fig. 6), followed by GEN, and FRA, suggesting a dominant westerly and northerly air flow. The recirculation of air masses in the Gulf of Genoa could accentuate the sea breeze and therefore more O$_3$ coming from NO$_x$ associated with shipping activities in the Mediterranean will be transported to the coastal and inland station.

The high Alps receptor region is less influenced by ALP emissions than the Po Valley, and is more influenced by remote sources (see Fig. 6). The increased contribution of O$_3$ from CEE, ITA and FRA to both exceedance and non-exceedance days in the high Alps receptor region compared with the Po Valley receptor region highlights the impact of the transboundary transport of O$_3$ and its precursors . Furthermore, the contribution of stratospheric as well as long-range sources was generally 6 % higher in this receptor region than in the Po Valley receptor region.

In GEN, the main remote source regions are FRA and CEE during the exceedance days and FRA and UKI during non-exceedance days (Fig. 6). Opposite to Po Valley, in GEN the model predicts fewer MDA8 O$_3$ exceedances days. Comparing the source contribution to both modelled and observed exceedances days, we noticed that the model underestimates O$_3$ concentrations associated with long-range transport and natural sources. Further, the model predicted higher levels of O$_3$ from CEE and FRA than observed. Underestimation of long-range transported O$_3$ into the GEN region in our model could be explained by the fact that the number of modelled MDA8 O$_3$ exceedances in GEN is half of the observed number of exceedances (Fig. 6).

This kind of analysis can be applied to improve our knowledge of the origin of O$_3$ precursors and their contribution to MDA8 O$_3$ health metrics. Hence, by using this tagging technique, policymakers can identify future actions required to control the NO$_x$ emissions at local and regional levels.

### 3.4 Tagged ozone precursor contributions to regulatory ozone metrics

In this section, we discuss the contribution of O$_3$ mixing ratios from diverse emissions sources and types to several metrics that quantify the O$_3$ exposures of humans and ecosystems. From modelled hourly mixing ratios of tagged O$_3$ sources and other global types, we have calculated different O$_3$ metrics, including non-cumulative (mean, MDA8, and the 95$^{th}$ percentile O$_3$) and cumulative (SOMO35, W126, and AOT40) metrics. We have chosen not to analyse the performance of the calculated cumulative metrics in comparison with measured values, as was done in previous work by Tong et al. (2009). Their work

showed that the poor performance of the cumulative metrics is closely related to the sensitivity of these metrics to the threshold values or weighting factors.

Figure 7 and Table S2 include the percentage of the contribution of different sources of emissions and other global types to total $O_3$ as calculated using health and vegetation metrics. The non-cumulative $O_3$ metrics employed in this study have
displayed similar patterns for most of the receptor regions. The contribution of local and European sources to the total $O_3$ mixing ratios have been low when we applied to mean $O_3$ metric and high when using $95^{th}$ percentile metric. These findings emphasize the importance of $O_3$ produced by local and neighbouring sources to the high end of the $O_3$ mixing ratio distribution.

Splitting the non-cumulative metrics into early (April-June) and late (July-September) simulation periods clearly illustrates that the European receptor regions are more prone to be influenced by intercontinental transport during the early period than
the late period. The contribution of intercontinentally transported $O_3$ to mean $O_3$ values in different receptor regions is higher during the early period and it spans between 22.8 % and 54.3 % of total $O_3$. In the late period it accounts for between 16 % and 48.9 % of total $O_3$ (see Fig. 7 and Table S2). Since $O_3$ associated with intercontinental transport comes, in this case, solely from boundary conditions, errors in boundary conditions affect the predicted mixing ratio of various chemical species, and, consequently, the contribution of overseas sources of $O_3$ to levels observed in Europe $O_3$ (Tang et al., 2007; Giordano et al.,
2015; Im et al., 2018).

The shorter lifespan of $O_3$ over remote ocean regions throughout the warm season, combined with synoptic conditions, has led to decreased levels of intercontinentally transported $O_3$ to Europe. Thus, for most receptor regions, the $O_3$ coming from Asia and the rest of the world was reduced by more than half when compared with the cold period. The $O_3$ mixing ratio from the stratosphere is, in general, 2.5 times higher in the cold season than in the warm season which is consistent with the findings
of a study by Butler et al. (2018) which showed that the stratospheric $O_3$ mixing ratio varies with altitude and its lifetime is influenced by season and latitude. The tagging technique also helps to quantify the impact of biogenic and biomass burning emissions of $NO_x$ on tropospheric $O_3$. The impact of biogenic $NO_x$ emissions on mean $O_3$ mixing ratios is between 3.3 % in Po Valley and 5.9 % in TCA in the early season, while during the late season it is between 5.4 % in Po Valley and 13.4 % in RBU. The biomass burning emissions account for variable percentages of mean $O_3$ mixing ratios. These span between 1.6 %
in ITA to 5.3 % in RBU during the early season, and between 3.8 % in Po Valley and 16.3 % in RBU during the late season. Natural sources do not usually vary greatly when different non-cumulative metrics are applied. An exception would be for the biomass burning emissions on RBU during the late season. Thus, BMB in RBU contributes to 16.3 %, 17.6 % and 28.8 % of the mean, MDA8 and $95^{th}$ percentile, respectively.

Even though the SOMO35 and AOT40 metrics are not accumulated over the same period (SOMO35 is accumulated over
the entire simulated period, and AOT40 metric is accumulated over the May-July period) and do not use same input data (daily MDA8 $O_3$ for SOMO35 vs daytime $O_3$ mixing ratios for AOT40), since they are based on threshold exceedances and are designed to measure exposure to high $O_3$ levels of humans (SOMO35) and vegetation (AOT40), there is a way to directly compare data from each metric type. As shown in Figure 7 and Table S2, the contribution of different sources of emissions and types as a proportion of total SOMO35 and AOT40 metrics is similar for most of the European receptor regions. Their spatial
distribution (not shown) is also comparable, with minimum values over the UK, NW Europe and Scandinavia and maximum

values over Italy, the Alps, south of Spain, east of Turkey and in the metropolitan area of Moscow, Russia. These results are consistent with previous studies performed by Aksoyoglu et al. (2014), and Anav et al. (2016). The overseas sources contribute similarly to SOMO35 and AOT40 indexes (usually less than 30 %) for most of the receptor regions used in this study. However, in UKI the overseas sources account for 32 % of AOT40 and 38 % of SOMO35, and in SCA they contribute to ~22 % of AOT40 and 30 % of SOMO35. This suggests that these metrics are more sensitive with respect to the $O_3$ mixing ratios from remote sources in areas having a low level of $O_3$ pollution. In the RBU receptor region, these indicators are sensitive to $O_3$ coming from biomass burning emissions (20 % of SOMO35 and 24 % of AOT40), whereas for the remaining receptor regions the contribution of natural sources to SOMO35 and AOT40 is similar. Local sources account for a range of ~12 % (SCA) to ~38 % (GEN) of these metrics. These data highlight the occurrence of increased $O_3$ production from local sources in comparison with northern European countries as well as large emissions of $NO_x$ in the GEN source region. Since the difference between AOT40 and SOMO35 is only a few percentage points, regardless of the receptor region, we were able to conclude that they behave similarly, according to thresholds used to define these metrics.

The tagging method allows a better understanding of the main precursor sources responsible for exceedances of regulatory $O_3$ metrics. This information can help to inform further modelling studies aimed at investigating the effects of emission reduction strategies, and ultimately inform air quality policy. For example, in the Po Valley receptor region, the modelled AOT40 is up to 3.4 times higher than the target limit given by EU legislation (on average 31218 $ppb - hours$). The observed and calculated AOT40 values depicted in Figure S3 exhibit the exceedance of target limits in Po Valley. $O_3$ coming from local sources can explain 35.0 % of this value (an average of 10909 $ppb - hours$). After local sources, the main European anthropogenic sources contributing to high AOT40 values in the Po Valley region are from FRA (6.6 %), GEN (7 %) and MBS (8.8 %) (Table S2). Generally, the $O_3$ mixing ratio and its precursors transported from other anthropogenic European sources into the Po Valley receptor regions account for ~39.5 %, while natural sources account for ~12.3 % and long-range transport accounts for ~13.4 % of the remaining AOT40 mixing ratios. Thus, to reach at least the target limit in the Po Valley receptor region, considerable emission reductions will still be needed, not only on a local scale but also on the European scale, especially within the MBS, GER, and FRA source regions.

Figure 7 also shows the percentage that different types of emissions and emission regions contribute to the W126 index. Interestingly, for most of the receptor regions, local $NO_x$ anthropogenic emissions cause the largest response in W126 values compared with the other cumulative metrics used in this study. Thus, local $NO_x$ explains from 10.9 % (0.1 $ppm - hours$) in SCA to more than 40 % of W126 in GEN (45.9 %; 2.48 $ppm - hours$, and Po Valley (45.4 %; 8.7 $ppm - hours$) of W126 index values calculated for each region. The effect of European transported plumes is also enhanced when using the W126 index compared with the other metrics for most of the downwind receptor regions. This behaviour is related to how these metrics have been defined. Due to its sigmoidal weighted formulation, as discussed in Westenbarger and Frisvold (1995), and Lapina et al. (2014), W126 includes all daytime values rather than $O_3$ levels above a certain threshold, as is done using SOMO35 and AOT40; therefore lower weighting factors of less than 0.5 are given to low $O_3$ values and weighting factors above 0.5 are given to $O_3$ values situated above the inflection point of 67 ppb.

The modelled mean AOT40 and W126 values in the Po Valley receptor region exceeded standards (26368 $ppb - hours$ for AOT40 and 28.9 $ppm - hours$ for W126) during the May-July 2010 period, and, as shown in Fig. 7 and Table S2, local sources are an important contributor to these metrics. To better understand why the W126 index is mainly influenced by local sources compared with the other cumulative metrics, we thoroughly compared AOT40 and W126 values for the Po Valley receptor region. As shown in Figure 8, a temporal series of hourly daylight values for mean $O_3$, W126 and AOT40 values averaged over the Po Valley receptor region are given. Since the W126 unit is $ppm - hours$, a more direct comparison with the W126 index would require values be expressed in $ppb - hours$. Further, all metrics showed a similar level of temporal variation in which they peaked in the first half of July. Also, whenever the averaged $O_3$ mixing ratio was lower than 60 $ppb$ (Fig. 8a), W126 value was lower than AOT40 (Fig. 8d). This way of acting was most probably due to the weighting factor being less than 0.3, and above this mixing ratio W126 tends to be higher than AOT40. This behaviour is closely linked to the definition of these metrics. If the $O_3$ mixing ratio is less than 40 $ppb$, W126 has a weighting factor lower than 0.03, while AOT40 has a weighting factor of 0. Above this threshold, AOT40 has a weighting factor of 1, while in the case of W126 only $O_3$ values higher than 100 $ppb$ have a weighting factor of 1. Due to the way these metrics are defined, predicted $O_3$ values in each grid cell are accounted for the W126, may not be accounted for the AOT40 index.

In addition, visual analysis of the time series also revealed that when the $O_3$ mixing ratios from local sources are ~20 $ppb$, these mixing ratios have a higher contribution to W126 than AOT40. To better understand this observation, we have further analysed the relationship between mean $O_3$ values from ALP sources ($O_3$-ALP) and the percent contribution of these $O_3$ tracers to mean $O_3$, W126, and AOT40 metrics. Figure 9 shows scatter plots for $O_3$-ALP that relate the contributions of these mixing ratios on mean $O_3$, W126, and AOT40. In addition, the linear regressions of Y vs X (Y=a*X+b) using all data sets have been applied. We saw that in general, high mean $O_3$-ALP mixing ratios have a higher contribution to W126 than to AOT40; this was also confirmed by the linear regression between $O_3$-ALP and W126 that yields a slope of 1.52 compared to a slope of 1.36 obtained when the linear regression was applied to AOT40 vs. $O_3$-ALP. Averaged $O_3$-ALP and mean $O_3$ as well as $O_3$-ALP and W126 were highly correlated (r=0.96, and r=0.93,respectively), while $O_3$-ALP and AOT40 are correlated more loosely (0.88). The high level of correlation between $O_3$-ALP and both mean $O_3$ and W126 could be related to the fact that these metrics account for all modelled values, whilst AOT40 considers only $O_3$ values above 40 $ppb$.

Extending this analysis to all receptor regions, we can explain why the W126 index is more sensitive to $O_3$ coming from local sources compared with the other cumulative metrics. In addition, W126 accentuates the contribution of BIO and BMB in RBU, TCA and SEE, most likely because the metric includes all daytime values, and not just those above a certain threshold. Thus, the use of W126 highlights the considerable impacts of BIO and BMB emissions on total $O_3$ mixing ratios throughout the summer and from burning vegetation that ultimately influence the extent to which $O_3$ causes damage to vegetation.

We have seen that the contribution of $NO_x$ to total $O_3$ varies depending on metrics and regions considered. Hence, the tagging method could help design different emission control strategies in specific source regions depending on which impacts need to be reduced in specific receptor regions.

## 4 Conclusions

Here, we implemented a new chemical mechanism within the WRF-Chem model to account for source attribution of $O_3$ from $NO_x$. We investigated the origin of surface $O_3$ using the "tagging" technique from April-September 2010, as well as the contribution of different sources to $O_3$ metrics, and their exceedance events.

Using tagged simulation from WRF-Chem, we show that the spatial distribution of simulated monthly mean MDA8 from tagged $O_3$ source regions and other global types throughout late spring, summer and early autumn of 2010. The contribution of different sources to $O_3$ production varies with season. We have identified intercontinental transported $O_3$ as an important contributor to the total $O_3$ mixing ratio, especially in the late spring and early autumn. During summer, however, the $O_3$ production is dominated by national and intra-European sources. We have also identified shipping activities in the Mediterranean Sea as an important source of $O_3$ for the IBE, ITA, SEE, and FRA peripheral maritime receptor regions. We also analysed the main sources of MDA8 $O_3$ over the Mediterranean Basin and we have identified the main factors that contribute to MDA8 $O_3$ mixing ratios to the greatest degree. These were mainly shipping activities and the localized contribution from the bordering countries.

To better understand the origin of MDA8 $O_3$ exceedances, we compared modelled and observed values of MDA8 $O_3$ concentration in the Po Valley, high Alps, Germany, and Benelux receptor regions. Throughout days exceeding the recommended thresholds of 120 $\mu$g, the contribution from local sources was ~41 %, 34 % and 38 % of modelled MDA8 $O_3$ for Po Valley, high Alps, and GEN, respectively. Throughout days not exceeding recommended thresholds, local emissions explain ~27 %, 16 % and 23% of modelled MDA8 $O_3$ for the Po Valley, high Alps, and GEN, respectively. Moreover, this tagging approach revealed that the main remote sources of MDA8 $O_3$ are MBS, GEN, and FRA for the Po Valley receptor region, and are FRA, CEE and UKI for Germany and Benelux receptor region. In addition, these analyses identified a persistently high contribution of transboundary sources to background $O_3$ concentration in the high Alps receptor region. Furthermore, by showing that the contribution of precursor sources to modelled $O_3$ target value exceedances in the GEN region is systematically different from the contribution of precursor sources to modelled $O_3$ when exceedances are observed but not modelled, we have identified a possible reason (underestimation of long-range transport) for the poor performance of our model with respect to reproducing the observed number of $O_3$ target value exceedances in the GEN region.

Through comparisons with different $O_3$ metrics, we quantified the impact of local vs. non-local $NO_x$ on $O_3$ production for each European receptor region. The comparison between mean, MDA8 and $95^{th}$ percentile $O_3$ metrics accentuate the importance of large contributions from different $NO_x$ to the high-end of the $O_3$ distribution. By analysing these metrics for two periods (April-June and July-September), we can clearly distinguish the contribution of different $NO_x$ to total $O_3$ mixing ratios in each region and throughout different times of the year. When we compare the cumulative metrics, we noticed that the SOMO35 and AOT40 indexes exhibit rather similar behaviour. Considering that these metrics are not calculated over the same period nor do they use same input data, the similar behaviour is likely due to the similar threshold values applied to define these metrics.

The use of the W126 index accentuates the importance of local emissions. To confirm this, we investigated the behaviour of modelled mean AOT40 and W126 values in the Po Valley receptor region. We noticed that when the local sources contribute to more than 20 ppb of the $O_3$ mixing ratios, these mixing ratios have a higher contribution to W126 than they do to AOT40 and determined that the difference was mostly due to the definition of W126 which takes into account all $O_3$ values, not only those that are above a certain threshold.

Overall, this study has identified local and remote factors that contribute to the MDA8 $O_3$ mixing ratio during several periods as well as within different $O_3$ metrics. Furthermore, the method applied here could be used to design improved emission control strategies depending on which impacts need to be reduced.

**Appendix A**

– chemics_init.F;

– module_input_chem_data.F;

– module_plumerise1.F and module_add_emiss_burn.F to account the source attribution of biomass burning emissions to $O_3$ concentration;

– module_emissions_anthropogenics.F to account for the impact of anthropogenic emissions on $O_3$ concentration;

– module_bioemi_megan2.F and module_data_mgn2mech.F to see the impact of biogenic emissions on $O_3$ concentration;

– module_lightning_nox_driver.F for lightning-generated nitrogen oxides

– Dry and wet deposition of tagged trace gases are treated by module_dep_simple.F and module_mozcart_wetscav.F, thus all tagged species have the same dry deposition velocities and wet removal rates with the corresponding non-tagged species;

– module_ftuv_driver.F to consider the photolytical reaction of the new packages;

– emissions_driver.F;

– chem_driver.F.

*Code and data availability.* The WRF-Chem model is publicly available on http://www2. mmm.ucar.edu/wrf/users/download/get_source.html. The modifications introduced and described in Section 2 are available online via ZENODO at https://doi.org/10.5281/zenodo.3501963. The model data can be provided upon request to the corresponding author.

*Author contributions.* AL and TB designed the research. AL adapted the automatic mechanism-rewriting and code-generation tools and in implemented into WRF-Chem source code. AL performed the model runs and subsequent analysis. AL wrote the pape with contribution from TB.

*Acknowledgements.* This work was hosted by IASS Potsdam, with financial support provided by the Federal Ministry of Education and Research of Germany (fBMBF) and the Ministry for Science, Research and Culture of the State of Brandenburg (MWFK). The authors would like to thank Kathleen Mar for helping with the emissions preprocessing as well as to Jane Coates for her help with some of the plots.

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

**Table 1.** List of tagged source regions and global source types

| Category | Acronym | List of countries or global source types |
|---|---|---|
| European source regions | MBS | Mediterranean, and Black Seas |
| | BNS | Baltic, and North Seas |
| | CEE | Central-East Europe includes East Austria, Hungary, Czech Republic, Slovakia, Estonia, Latvia, Lithuania, Poland |
| | ALP | High Alps (West Austria, Switzerland, and North Italy) and Po Valley |
| | ITA | South Italy, and Malta |
| | SEE | South-East Europe includes Bulgaria, Romania, Moldavia, Albania, Slovenia, Croatia, Serbia, Montenegro, Macedonia, Greece, and Cyprus |
| | IBE | Iberia includes Spain, and Portugal |
| | UKI | United Kingdom, and Ireland |
| | GEN | Germany, Belgium, Netherlands, and Luxembourg |
| | SCA | Scandinavia (Finland, Norway, Sweden), Denmark, and Island |
| | FRA | France |
| | RBU | Russia, Belarus, and Ukraine |
| | TCA | Turkey, Azerbaijan, Armenia, and Georgia |
| HTAP2 source regions | ASI | Chemical boundary condition of modelled species coming from Asia |
| | NAF | Chemical boundary condition of modelled species coming from North-Africa |
| | NAM | Chemical boundary condition of modelled species coming from North-America |
| | OCN | Chemical boundary condition of modelled species coming from shipping activities |
| | RST | Chemical boundary condition of modelled species coming from rest of the world |
| Global source types | BIO | biogenic |
| | BMB | biomass burning |
| | LGT | lightning |
| | STR | stratospheric O3 |

**Table 2.** Observed mean and simulation summary statistics for meteorological parameters. The normalized mean bias (NMB) and correlation coefficient (R) are calculated between simulated and observed meteorological observation from GWO during April – September 2010 period

| Variable | Observed | | | Modelled mean | | | NMB (%) | R |
|---|---|---|---|---|---|---|---|---|
| | min | mean | max | min | mean | max | | |
| MSLP (hPa) | 1000.96 | 1014.3 | 1022.06 | 969.05 | 1014.5 | 1039.03 | -0 | 0.98 |
| T2M (°C) | -17.14 | 14.99 | 32.10 | -22.50 | 14.76 | 43.45 | -3 | 0.91 |
| WS10M (m/s) | 0.36 | 3.37 | 10.83 | 0.00 | 3.59 | 20.41 | 8 | 0.65 |
| WD10M (°) | 0 | 190 | 360 | 31.91 | 216 | 318 | 13 | 0.47 |

**Table 3.** Observed mean and simulation summary statistics for MDA8 $O_3$ concentrations ($\mu g/m^{-3}$) at rural background sites. The normalized mean bias (NMB) and correlation coefficient (R) are calculated between simulated and observed $O_3$ concentrations from the AirBase dataset during April – September 2010 period.

| Analyzed period | Observed | | | Modelled | | | NMB (%) | R |
|---|---|---|---|---|---|---|---|---|
| | min | mean | max | min | mean | max | | |
| April | 52.5 | 97.0 | 140.8 | 36.5 | 90.8 | 134.5 | -6.3 | 0.58 |
| May | 41.0 | 87.9 | 143.0 | 28.0 | 83.2 | 124.6 | -5.4 | 0.62 |
| June | 44.2 | 96.2 | 162.3 | 32.0 | 89.7 | 132.6 | -6.8 | 0.71 |
| July | 43.8 | 97.0 | 178.2 | 26.0 | 90.8 | 147.7 | -6.3 | 0.58 |
| August | 40.3 | 87.5 | 145.2 | 27.3 | 82.6 | 130.8 | -5.6 | 0.65 |
| September | 33.4 | 77.5 | 135.4 | 26.5 | 81.1 | 129.6 | 4.6 | 0.63 |
| Total | 40.5 | 90.5 | 160.5 | 28.4 | 86.3 | 135.9 | -5.2 | 0.69 |

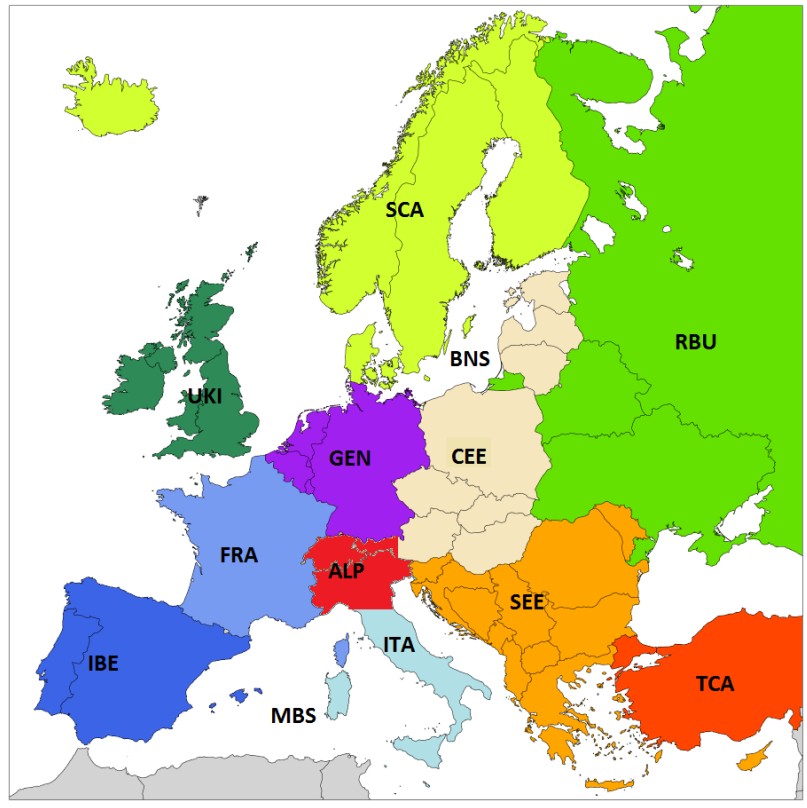

**Figure 1.** Tagged European source regions

## MDA8 O3 (ppb) - April-May 2010

**Figure 2.** Contribution to MDA8 O$_3$ (ppb) of each O$_3$ source region and global source type for the April-May 2010 period

# MDA8 O3 (ppb) - June-August 2010

**Figure 3.** Same as Fig. 2, but for the June-August 2010 period

# MDA8 O3 (ppb) - September 2010

**Figure 4.** Same as Fig. 2, but for September 2010

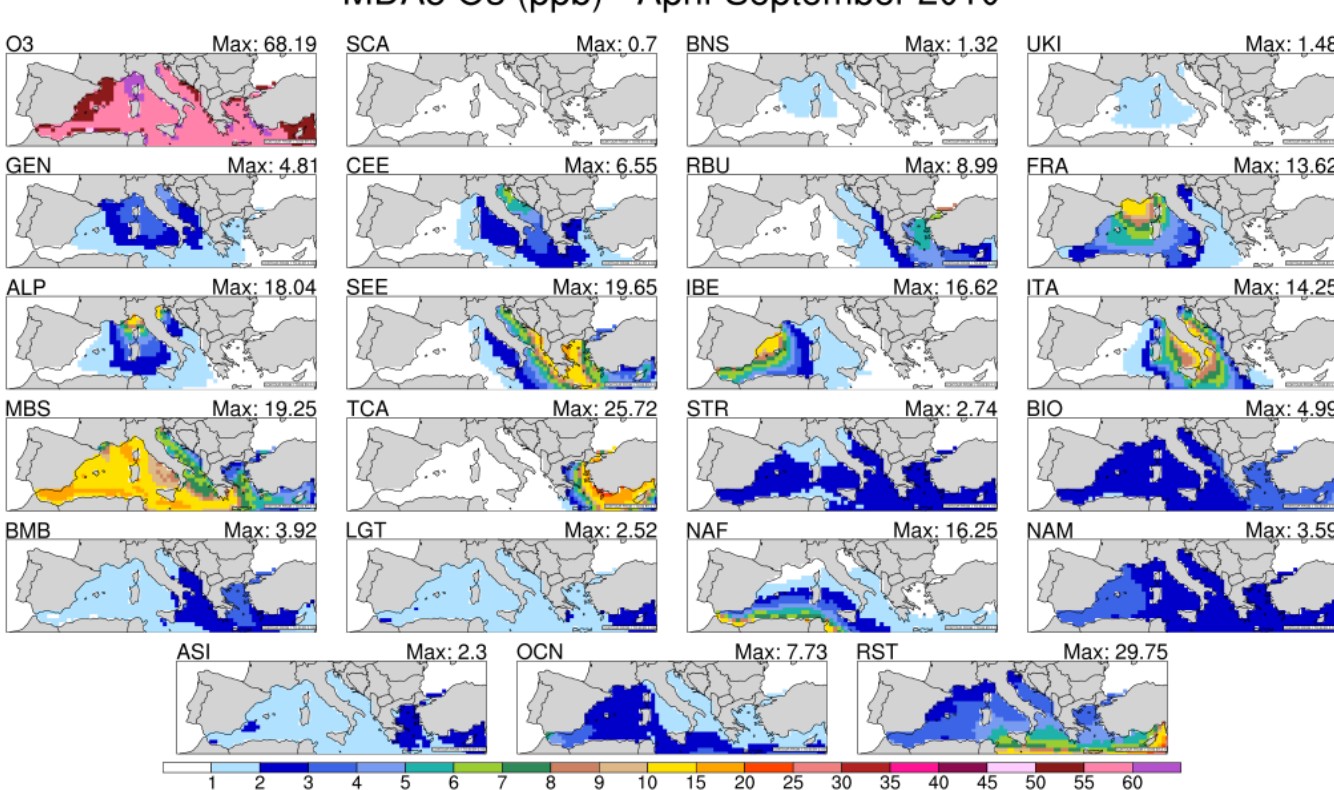

**Figure 5.** Average MDA8 O₃ mixing ratio (upper left panel) and contribution of each tagged O₃ source region and global source type over the Mediterranean Sea for the April-September 2010 period. The unit is ppb

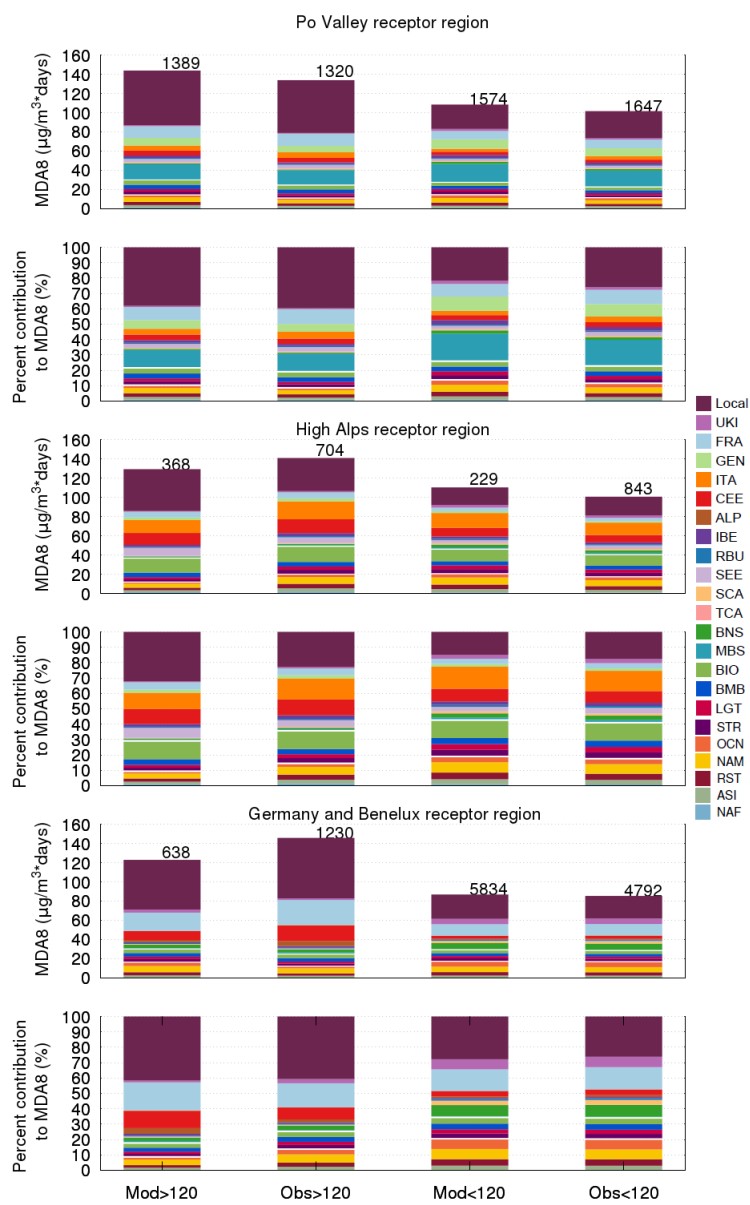

**Figure 6.** Mean modelled and observed MDA8 O$_3$ mixing ratio filtered by a threshold of 120 $\mu$ g m$^{-3}$ for Po Valley, (top panel) high Alps (third from top panel) and GEN (fifth from top panel) and percent contribution to MDA8 O$_3$ from different emissions sources and types for Po Valley (second panel), high Alps (fourth) and GEN (bottom panel) during April-September 2010 period. In each case the contributions of tagged sources to the total O$_3$ are shown. The tagged contributions of local and other European sources, HTAP2 source regions and other global source types to observed O$_3$ are obtained by scaling the observed O$_3$ by the relative contributions of these tagged sources to modelled O$_3$. The total number of exceedances (and non-exceedances) of the MDA8 O$_3$ target value is indicated at the top of each column

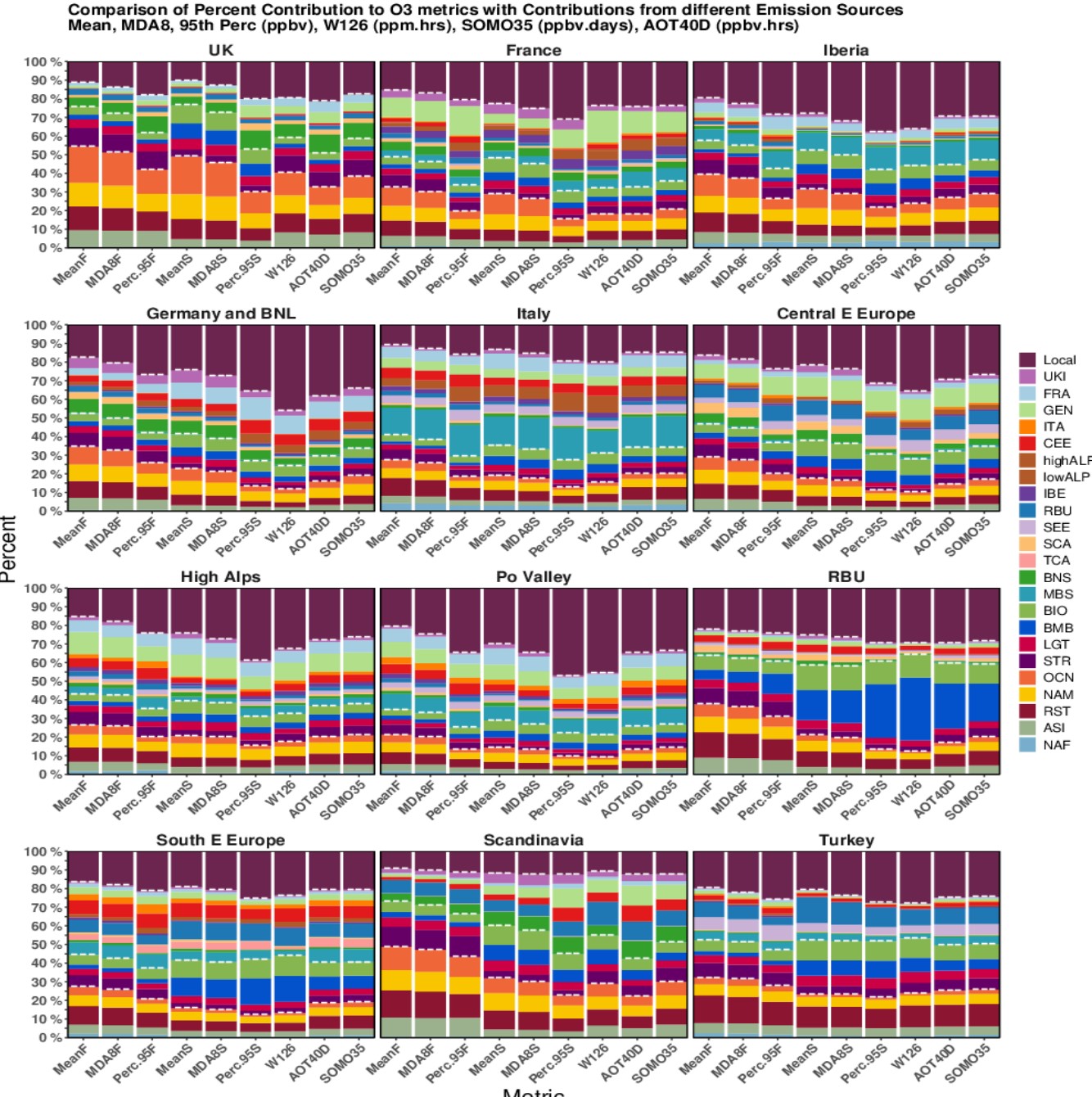

**Figure 7.** Comparison of percent contribution of local and other European sources, HTAP2 source regions and other global source types to different to O₃ metric. The metrics analysed are mean, MDA8 and $95^{th}$ percentile (ppb) for the early "F" and late "S" simulation period, W126 (ppm − hours), SOMO35 and AOT40 (ppb − hours). The white dashed lines on each panel separate different categories (intercontinental transport, other global source types, and local and other European sources)

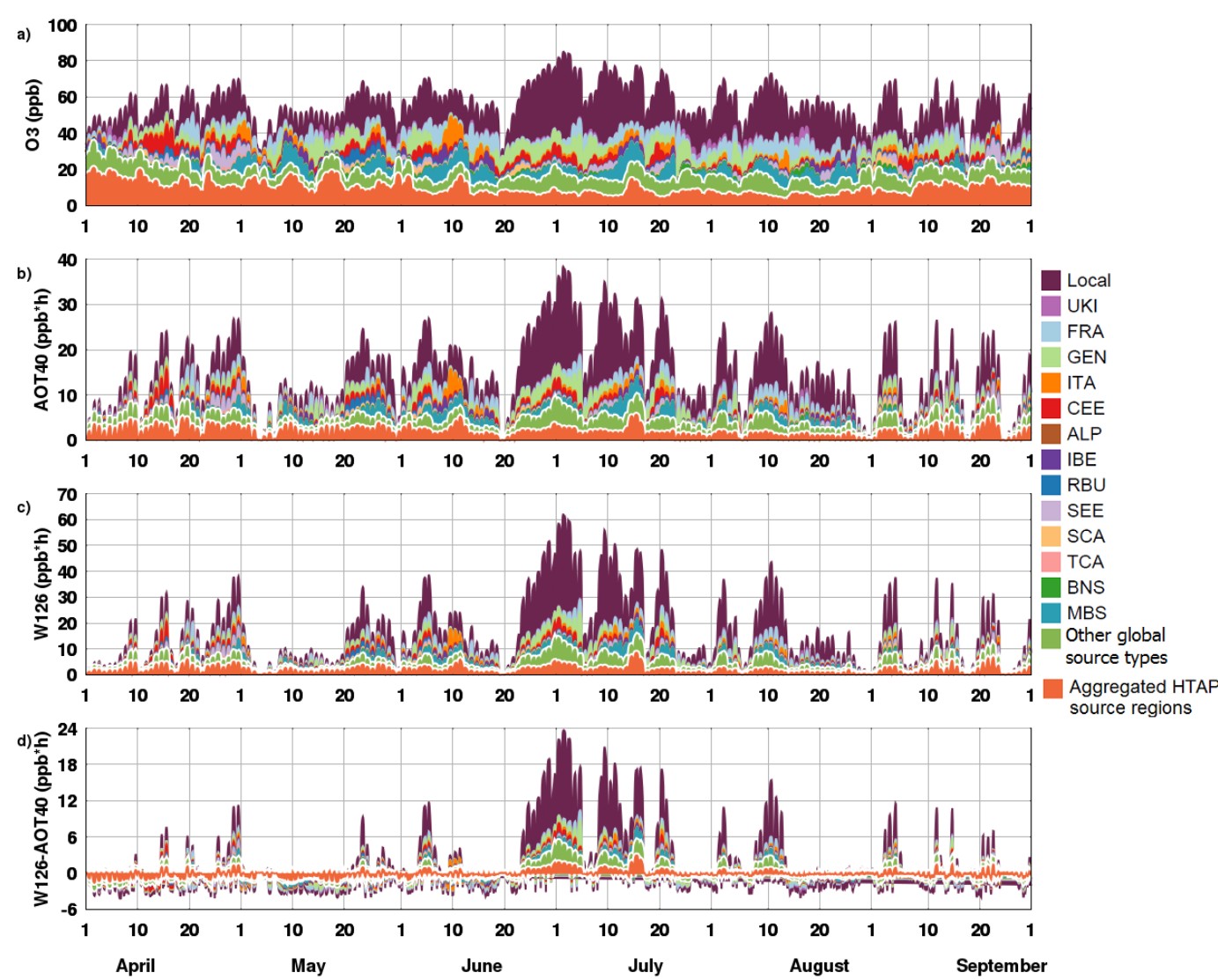

**Figure 8.** April-September 2010 time series of daytime a) hourly $O_3$ (ppb), b) hourly AOT40 index $(ppb - hours)$, c) hourly W126 index $(ppb - hours)$ and d) differences between W126 and AOT40 indexes $(ppb - hours)$ averaged over Po Valley receptor region. The colour bars indicate the $O_3$ source categories - aggregated HTAP2 regions (ASI, NAM, NAF, OCN and RST), the other global source types (STR, LGT, BMB and BIO) and the European source regions. The white dashed lines on each panel separate different categories (intercontinental transport, other global source types, and local and other European sources).

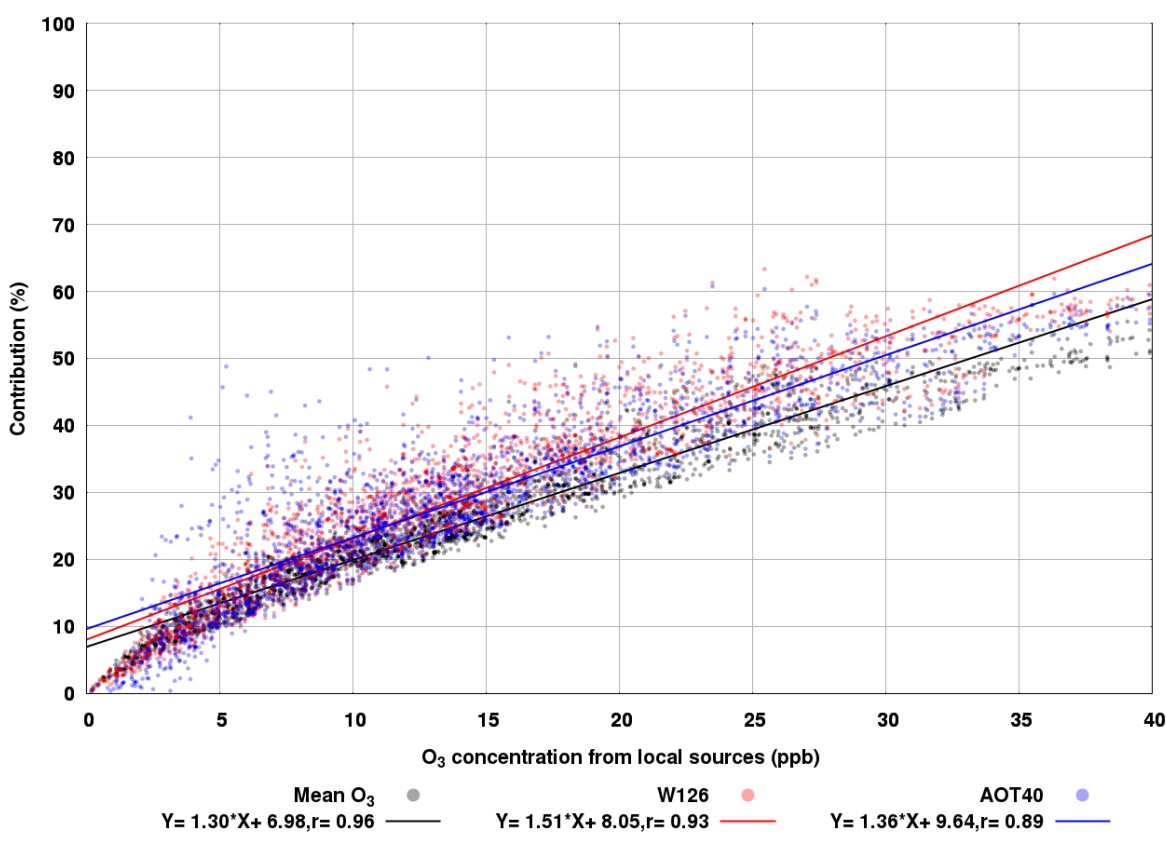

**Figure 9.** Scatter plots showing the ozone concentration from local sources versus the contribution to Mean $O_3$ (black dots), W126 (red dots) and AOT40 (blue dots). The solid lines are the lines of best fit.