# Peer review of "Source attribution of European surface O3 using a tagged O3 mechanism"

_Atmospheric Chemistry and Physics, 2019_

## Referee Comment (RC1) · Anonymous Referee #3 · 23 May 2019

This is a very nice analysis that provides a lot of useful information and insight regarding the sources of ozone across Europe. The material is appropriate for ACP and I could recommend the paper for publication after it is revised according to my comments below.

General comments: 1) My only concern about the analysis is with regards to the ALPS region. This region combines the high elevations of the Alps (strong influence from long range transport of European and intercontinental origin) with the low elevations of the Po Valley, which is shielded from long-range transport by the Alps and experiences localized and intense air pollution episodes. Given the high variability of source regions and the high variability of local emissions, I don't think that any clear conclusions can be drawn for this region. The authors need to split this region into two parts: 1) the

Po Valley; and 2) regions above 1500 m elevation. Then we should see that the high elevations have the greatest impact from long-range transport and the low elevations have the strongest impact from local emissions.

2) In general the standard of English is fairly good but it needs a lot of polishing. There are too many instances of grammatical errors or awkward phrasing for me to list individually, but here are a few examples from the first paragraph of the Introduction: "The World Health Organization air quality guideline report that high O3 concentrations can cause damages to humans and vegetation" "Moreover, it has been shown that tropospheric O3 also affects radiative forcing and therefore contributing to climate change." "To maintain a good air quality and understand O3's response to climate change, it is important to understand the contribution of different sources of its precursors to the tropospheric O3 concentration." Another example is this sentence in the Conclusions, which is difficult to understand: "Thus, we have seen that during the exceedances days, the contribution from local sources sources is ∼45 % and 38 % of modeled MDA8 O3, whilst during nonexceedances values is ∼32 % and 2 3% for ALP, respectively GEN."

Specific Comments:

Page 2, line 5 Here you state: "It has been shown that the background O3 concentrations have increased during the last several decades due to the increase of overall global anthropogenic emissions of O3 precursors (HTAP, 2010)" What is meant by background? HTAP uses the term global background to refer to natural ozone that would exist in the absence of anthropogenic emissions. This quantity cannot be measured but must be calculated by global models. Do you really mean to say that the global natural background has increased? Or do you mean that average observed global ozone has increased? According to the extensive review conducted by TOAR-Climate [Gaudel et al., 2018] the current in situ ozone monitoring network is insufficient to quantify ozone changes on the global or hemispheric scale; the available satellite products disagree on the trend, with some showing increases and some showing decreases.

Line 6 Here you want to demonstrate that ozone air pollution is a current issue, but you only provide one reference that is now 11 years old and only applies to China. Please find additional references that are more current and cover other regions of the world. TOAR has recently published three papers that report present-day ozone using metrics relevant to human health [Fleming et al., 2018], vegetation [Mills et al., 2018] and climate [Gaudel et al., 2018]. These papers also provide up-to-date reviews of the literature concerning the impacts of ozone on humans, vegetation and climate. These papers would be helpful for the Introduction.

In general the standard of English is fairly good but it needs some polishing, for example the following phrases in the first paragraph of the Introduction need more work: "The World Health Organization air quality guideline report that high O3 concentrations can cause damages to humans and vegetation" "Moreover, it has been shown that tropospheric O3 also affects radiative forcing and therefore contributing to climate change. "To maintain a good air quality and understand O3's response to climate change, it is important to understand the contribution of different sources of its precursors to the tropospheric O3 concentration." Another example is this sentence in the Conclusions, which is difficult to understand: "Thus, we have seen that during the exceedances days, the contribution from local sources sources is ∼45 % and 38 % of modeled MDA8 O3, whilst during nonexceedances values is ∼32 % and 2 3% for ALP, respectively GEN."

Page 3, Line 21 Lefohn and Musselman stated that the W126 index ". . . would provide a more appropriate target for air quality management programs designed to reduce emissions from anthropogenic sources contributing to O3 formation". I'm not sure why this quote is provided. It seems to imply that W126 is a better metric then AOT40, but there is no agreement among the scientists who study the impacts of ozone on vegetation as to which metric is best. W126 was developed for a few limited crops and trees and is not necessarily applicable to other types of vegetation. For example, Harmens et al. [2018] found that a particular type of wheat is not sensitive to ozone peaks, which means W126 is not the best metric. Mills et al. [2018] give an overview

of various vegetation metrics and they point out that the flux-based methods are the most accurate, but TOAR did not include these methods because their calculation on the global scale is not yet feasible.

Page 4, Line 4 "Butler et al. (2018) describes in detail" should be "Butler et al. (2018) describe in detail" Because Butler et al. indicates many people, not one.

Page 4, line 18 I don't understand this phrase: "has been modified to enable to model capacity to be used"

Page 4, Line 19 Should this sentence be a part of the preceding paragraph?

Page 5 line 31 Here and throughout you need to be consistent with regards to the term, ozone concentration. "Ozone concentration" is appropriate when using units of $\mu$g m$-3$, but when using units of ppb, the correct term is "ozone mixing ratio"

In many places in the paper the term "NOx precursors" is used. This seems to be a redundant phase. Just define NOx as a precursor gas at the beginning of the paper and then afterwards just use "NOx" as it will be clear that it is a well-known precursor.

Page 8, line 18 "In all receptor regions, the MDA8 O3 concentration is dominated by O3 produced by remote anthropogenic precursors" I think this is overstated because it gives the impression that far more than half of the ozone is from remote anthropogenic precursors. But the range is 30-53%. Please re-phrase this sentence.

Page 8 line 25 This sentence seems out of place and it should appear before the discussion of the stratospheric source.

Page 8 line 29 "Arabic Saudi peninsula" should be Arabian Peninsula

Page 9 line 11 This statement is not quite correct: "Another consequence of enhanced photochemical activity during the summer season is the reduction of stratospheric O3" It should say: "Another consequence of enhanced photochemical activity during the summer season is that it reduces the relative influence of stratospheric O3" But in

addition to changing the relative impact, do the absolute values of stratospheric ozone also diminish in summer, relative to spring?

Page 9 line 31 Is OCN in these studies defined in the same way as in your study? If not then OCN should not be used when referencing the other studies.

Page 12, line 34 I don't understand what is meant by "efficiently complementing". I think you are trying to say that they behave similarly.

Page 13, line 21 Here you say that W126 is more sensitive to local emissions because it does not include a threshold. But, in addition, isn't it also more sensitive to local emissions because this metric places much more weight on the high ozone values, and high ozone is likely to be produced under hot stagnant conditions when local emissions are more important?

References:

Fleming, Z. L., R. M. Doherty et al. (2018), Tropospheric Ozone Assessment Report: Present-day ozone distribution and trends relevant to human health, Elem Sci Anth, 6(1):12, DOI: https://doi.org/10.1525/elementa.273

Gaudel, A., et al. (2018), Tropospheric Ozone Assessment Report: Present-day distribution and trends of tropospheric ozone relevant to climate and global atmospheric chemistry model evaluation, Elem Sci Anth, 6(1):39, DOI: https://doi.org/10.1525/elementa.291

Harmens, H., Hayes, F., Mills, G., Sharps, K., Osborne, S. and Pleijel, H., 2018. Wheat yield responses to stomatal uptake of ozone: Peak vs rising background ozone conditions. Atmospheric Environment, 173, pp.1-5.

Mills, G, et al. (2018), Tropospheric Ozone Assessment Report: Present-day tropospheric ozone distribution and trends relevant to vegetation. Elem Sci Anth, 6(1):47, DOI: https://doi.org/10.1525/elementa.302

---

## Referee Comment (RC2) · Anonymous Referee #2 · 27 Jul 2019

This is a well-written paper that provides a unique analysis of all of the source contributions to surface ozone in Europe over the spring to autumn months in 2010. By creating additional tracers of NO, NO2 and their reservoirs to the chemical mechanism, the sources of ozone can be quantified without perturbing emissions (which changes the chemistry), as is frequently done for such analyses. This paper discusses in a comprehensive and clear manner the various contributions and their seasonal variation. A nice analysis of standard metrics (MDA8, W126, AOT40 and SOMO35) is also presented. I recommend publication after addressing the minor comments outlined below.

1. There are numerous grammar errors (missing articles, plural/singular mismatch, etc.) that should be corrected.

[Figure]

2. In Section 3.1 more should be said about how similar this simulation is to the one evaluated in Mar et al. (2016) to assure the reader that the results of that evaluation are really relevant here.

3. The table captions in the supplement should more clearly explain the contents (units, etc.).

---

## Author Comment (AC1) · 4 Sep 2019

Please refer to the supplementary pdf file.

Please also note the supplement to this comment:
https://www.atmos-chem-phys-discuss.net/acp-2019-225/acp-2019-225-AC1-supplement.pdf

---

## Author Response (AR1)

**Reply to the reviewers' comments on the ACPD manuscript "Source attribution of European surface O3 using a tagged O3 mechanism" by Aurelia Lupaşcu and Tim Butler**

Below we address the comments of the reviewers and questions raised during the open discussion of the manuscript "Source attribution of European surface O3 using a tagged O3 mechanism". We thank to the two Anonymous Reviewers for the time and effort reviewing the manuscript. We believe it has improved thanks to their comments. We have listed all reviewers' comments below and our answers are provided in blue. We also append a "track-changes" version of the revised manuscript as an appendix with all changes to the manuscript highlighted.

**Response to Anonymous Referee #2 comments**

This is a well-written paper that provides a unique analysis of all of the source contributions to surface ozone in Europe over the spring to autumn months in 2010. By creating additional tracers of NO, NO2 and their reservoirs to the chemical mechanism, the sources of ozone can be quantified without perturbing emissions (which changes the chemistry), as is frequently done for such analyses. This paper discusses in a comprehensive and clear manner the various contributions and their seasonal variation. A nice analysis of standard metrics (MDA8, W126, AOT40 and SOMO35) is also presented. I recommend publication after addressing the minor comments outlined below.

We thank the reviewer for their positive comments. Below you can find our point-by-point responses to all the issues raised.

1. There are numerous grammar errors (missing articles, plural/singular mismatch,etc.) that should be corrected.

Thank you for the useful feedback. As suggested by both reviewers, the manuscript has been carefully revised and the grammatical, linguistic and spelling mistakes in the manuscript have been carefully checked and corrected.

2. In Section 3.1 more should be said about how similar this simulation is to the one evaluated in Mar et al. (2016) to assure the reader that the results of that evaluation are really relevant here.

Following the reviewer's suggestion, we updated the sentence as follows "An extensive evaluation of WRF-Chem using the MOZART chemical mechanism to predict long term meteorological data and O3 levels has been presented previously (Mar et al., 2016). The main differences between the set-up up used in this study and the model described by Mar et

al. (2016) include differences between the versions of the model used (3.7.1 vs. 3.5.1, respectively), horizontal resolutions (50kmx50km vs. 45x45km, respectively), microphysics (Morrison vs. Lin, respectively) and cumulus schemes (Grell-Freitas vs. Grell 3-D, respectively), simulation years (2010 vs. 2007, respectively), anthropogenic emissions inventory (TNO-MACC III vs. TNO-MACC II, respectively), and chemical input and boundary conditions (extended CAM-Chem version 1.2 with MOZART-4 vs. MOZART-4/GEOS-5 simulations found at http://www.acom.ucar.edu/wrfchem/mozart.shtml, respectively)."

3. The table captions in the supplement should more clearly explain the contents (units,etc.)
Following the reviewer's suggestion, we updated the Tables caption as it follows: "Table 1. Percent contribution of local, European, long-range transported (LRT) and natural emissions sources to MDA8 O3 (ppb) at each receptor region during late spring, summer and early autumn 2010" and "Table 2. Percent contribution of different emission sources and types to total O3 at each receptor region as calculated for health and vegetation metrics. The metrics analysed are mean, MDA8 and 95th percentile (ppb) for the early "F" and late "S" simulation period, W126 (ppm-hours), SOMO35 and AOT40 (ppb-hours)".

**Response to Anonymous Referee #3 comments**
This is a very nice analysis that provides a lot of useful information and insight regarding the sources of ozone across Europe. The material is appropriate for ACP and I could recommend the paper for publication after it is revised according to my comments below.
We thank the reviewer for their constructive and valuable comments. Following the Reviewer's suggestion, we revised the paper and addressed all the concerns raised by providing response to individual comments below. Our responses are in blue.

General comments: 1) My only concern about the analysis is with regards to the ALPS region. This region combines the high elevations of the Alps (strong influence from long range transport of European and intercontinental origin) with the low elevations of the Po Valley, which is shielded from long-range transport by the Alps and experiences localized and intense air pollution episodes. Given the high variability of source regions and the high variability of local emissions, I don't think that any clear conclusions can be drawn for this region. The authors need to split this region into two parts: 1) the Po Valley; and 2) regions above 1500 m elevation. Then we should see that the high elevations have the greatest impact from long-range transport and the low elevations have the strongest impact from local emissions.

Following the recommendation of the reviewer, we have split the ALP source region into two receptor regions: the Po Valley and the high Alps. The manuscript was revised to account for these regions and the discussion was updated accordingly in the following sections:

- 2.2 (Experimental setup) by adding the following sentences "Except for ALP, the source regions within the European domain are identical to receptor regions. Given the complex topography of the ALP source region, we split this region into two receptor regions: the Po Valley region and the high Alps (regions above 1500 m elevation)".

- 3.2 (Contribution of tagged precursor sources to the MDA8 O3 concentration) by recalculating the contribution of O3 coming from different sources and types to total O3 mixing ratio in the Po Valley and the high Alps receptor regions. We noticed that the contribution of different sources to the total MDA8 O3 mixing ratio in the Po Valley and the (old) ALP receptor regions give fairly similar results. Therefore, the text has been updated by using the newly calculated contributions in the Po Valley receptor region instead of ALP.

- 3.3 (Tagged ozone precursor contributions to exceedances of MDA8 target values – case study) by updating the manuscript as follows: "The relative contribution of emissions from different source regions to modelled MDA8 O3 and to observed MDA8 O3 values, after being scaled to account for the contribution of modelled sources of O3 types is generally similar for Po Valley and GEN receptor regions (see Fig. 6). In the Po Valley, we can pinpoint the main remote contributor as being MBS (see Fig. 6), followed by GEN, and FRA, suggesting a dominant westerly and northerly air flow. The recirculation of air masses in the Gulf of Genoa could accentuate the sea breeze and therefore more O3 coming from NOx associated with shipping activities in the Mediterranean will be transported to the coastal and inland station.  The high Alps receptor region is less influenced by ALP emissions than the Po Valley, and is more influenced by remote sources (see Fig. 6). The increased contribution of O3 from CEN, ITA and FRA to both exceedance and non-exceedance days in the high Alps receptor region compared with the Po Valley receptor region highlights the impact of the transboundary transport of O3 and its precursors. Furthermore, the contribution of stratospheric as well as long-range sources was generally 6 % higher in this receptor region than in the Po Valley receptor region."

- 3.4 (Tagged ozone precursor contributions to regulatory ozone metrics). We have recalculated the contribution of O3 coming from different sources and types to diverse health and vegetation metrics in the Po Valley and the high Alps receptor regions. Thus, in the revised manuscript, the comparison of different ozone metrics highlights the modelled values in the Po Valley instead of the ALP receptor region.

- We have updated the discussion in the Conclusion as follows: "To better understand the origin of MDA8 O3 exceedances, we compared modelled and observed values of MDA8 O3 concentration in the Po Valley, high Alps, Germany, and Benelux receptor regions. Throughout days exceeding the recommended thresholds of 120 µg/m$^{-3}$, the contribution from local sources was ~41 %, 34 % and 38 % of modelled MDA8 O3 for Po Valley, high Alps, and GEN, respectively. Throughout days not exceeding recommended thresholds, local emissions explain ~27 %, 16 % and 23% of modelled MDA8 O3 for the Po Valley, high Alps, and GEN, respectively. Moreover, this tagging approach revealed that the main remote sources of MDA8 O3 are MBS, GEN, and FRA for the Po Valley receptor region, and are FRA, CEN and UKI for Germany and Benelux receptor region. In addition, these analyses identified a persistently high contribution of transboundary sources to background O3 concentration in the high Alps receptor region".

We have also updated the tables in the Supplementary material accordingly. The details of the changed text could be also seen in the tracked changes version of the manuscript, appended here.

2) In general the standard of English is fairly good but it needs a lot of polishing. There are too many instances of grammatical errors or awkward phrasing for me to list individually, but here are a few examples from the first paragraph of the Introduction: "The World Health Organization air quality guideline report that high O3 concentrations can cause damages to humans and vegetation" "Moreover, it has been shown that tropospheric O3 also affects radiative forcing and therefore contributing to climate change." "To maintain a good air quality and understand O3's response to climate change, it is important to understand the contribution of different sources of its precursors to the tropospheric O3 concentration." Another example is this sentence in the Conclusions, which is difficult to understand: "Thus, we have seen that during the exceedances days, the contribution from local sources sources is~45 % and 38 % of modeled MDA8 O3, whilst during nonexceedances values is~32 % and 2 3% for ALP, respectively GEN."

Thank you for your kind suggestion. The manuscript has been carefully revised and the grammatical, linguistic and spelling mistakes in the manuscript have been carefully checked and corrected. Following the reviewer's suggestion, we have updated the introduction as follows "Ground-level O3 is an important air pollutant that damages human health (Fleming et al., 2018) and vegetation (Mills et al., 2018). It also affects the radiative forcing (e.g. Ramaswamy et al., 2001; Stevenson et al., 5 2013), and therefore contributes to climate change. Impacts of O3 on human health are associated with lung disease, chronic disease, and death from respiratory ailments. To protect human populations from exposure to high

levels of O3, the World Health Organization (WHO, 2006, 2017) recommended an air quality guideline for ozone in which the maximum daily average 8-h (MDA8) for O3 should not exceed 100 µg/m$^{-3}$. The European Environmental Agency (EEA, 2017a) reported that the EU long-term objective concentration of 120 µg/m$^{-3}$ is often exceeded and that more than 90 % of the urban population of the European Union was exposed to O3 levels higher than the stricter recommendation set by the WHO". "To improve the air quality in certain areas, it is important to know the extent to which different precursors (NOx and VOCs) contribute to tropospheric O3 concentrations".

We realize that the sentence in the Conclusions creates confusion, therefore, we have rephrased this as follows: "To better understand the origin of MDA8 O3 exceedances, we compared modelled and observed values of MDA8 O3 concentration in the Po Valley, high Alps, Germany, and Benelux receptor regions. Throughout days exceeding the recommended thresholds of 120 µg/m$^{-3}$, the contribution from local sources was ~41 %, 34 % and 38 % of modelled MDA8 O3 for Po Valley, high Alps, and GEN, respectively. Throughout days not exceeding recommended thresholds, local emissions explain ~27 %, 16 % and 23% of modelled MDA8 O3 for the Po Valley, high Alps, and GEN, respectively."

Specific Comments:

Page 2, line 5 Here you state: "It has been shown that the background O3 concentrations have increased during the last several decades due to the increase of overall global anthropogenic emissions of O3 precursors (HTAP, 2010)" What is meant by background? HTAP uses the term global background to refer to natural ozone that would exist in the absence of anthropogenic emissions. This quantity cannot be measured but must be calculated by global models. Do you really mean to say that the global natural background has increased? Or do you mean that average observed global ozone has increased? According to the extensive review conducted by TOAR-Climate [Gaudel et al., 2018] the current in situ ozone monitoring network is insufficient to quantify ozone changes on the global or hemispheric scale; the available satellite products disagree on the trend, with some showing increases and some showing decreases.

We meant the surface ozone without contribution from local anthropogenic sources, which should more correctly be referred to as the "baseline" ozone. The text has been changed accordingly as follows: "A 2010 report from HTAP (HTAP, 2010) shows that the observed baseline O3 concentrations (concentrations without the contribution from local anthropogenic emissions) have increased throughout the last several decades since overall global anthropogenic emissions of O3 precursors have increased. However, a more recent study by Gaudel et al. (2018) has established that the global surface O3 trends exhibit high variability, and depend on several factors such as season, region, elevation and proximity to fresh

ozone precursor emissions. However, since the network capable of monitoring ozone levels is sparse, it is difficult to quantify the O3 changes on a global scale. Satellite-derived O3 measurements can be used to quantify changing levels of O3, but Gaudel et al. (2018) showed that these products are not capable of quantifying significant trends."

Line 6 Here you want to demonstrate that ozone air pollution is a current issue, but you only provide one reference that is now 11 years old and only applies to China. Please find additional references that are more current and cover other regions of the world. TOAR has recently published three papers that report present-day ozone using metrics relevant to human health [Fleming et al., 2018], vegetation [Mills et al., 2018] and climate [Gaudel et al., 2018]. These papers also provide up-to-date reviews of the literature concerning the impacts of ozone on humans, vegetation and climate. These papers would be helpful for the Introduction.

Following the reviewer's suggestion, we revised the sentence as it follows: "Surface O3 pollution due to urbanization and motorization processes are serious challenges for large cities (e.g. Chan and Yao, 2008; Folberth et al., 2015; Li et al., 2017, 2019). Paoletti et al. (2014) showed that in Europe and the United States of America, the average O3 concentration in the cities has increased at a faster rate than those observed in rural areas. Fleming et al. (2018) showed that the 4th highest daily maximum 8-hour O3 (4MDA8) are more ubiquitous at urban sites than at non-urban sites. This leads to a worsening of general air quality that, ultimately, affects human health and ecosystems (Paoletti et al., 2014; Monks et al., 2015; WHO, 2017; Fleming et al., 2018; Mills et al., 2018)."

In general the standard of English is fairly good but it needs some polishing, for example the following phrases in the first paragraph of the Introduction need more work: "The World Health Organization air quality guideline report that high O3 concentrations can cause damages to humans and vegetation" "Moreover, it has been shown that tropospheric O3 also affects radiative forcing and therefore contributing to climate change" To maintain a good air quality and understand O3's response to climate change, it is important to understand the contribution of different sources of its precursors to the tropospheric O3 concentration." Another example is this sentence in the Conclusions, which is difficult to understand: "Thus, we have seen that during the exceedances days, the contribution from local sources sources is ~45 % and 38 % of modeled MDA8 O3,whilst during nonexceedances values is~32 % and 2 3% for ALP, respectively GEN."

The reviewer has apparently made a duplicate comment since s/he had referred to these phrases in the "General comments" section, which we have already responded to.

Page 3, Line 21 Lefohn and Musselman stated that the W126 index ". . . would provide a more appropriate target for air quality management programs designed to reduce emissions from anthropogenic sources contributing to O3 formation". I'm not sure why this quote is provided. It seems to imply that W126 is a better metric then AOT40, but there is no agreement among the scientists who study the impacts of ozone on vegetation as to which metric is best. W126 was developed for a few limited crops and trees and is not necessarily applicable to other types of vegetation. For example, Harmens et al. [2018] found that a particular type of wheat is not sensitive to ozone peaks, which means W126 is not the best metric. Mills et al. [2018] give an overview of various vegetation metrics and they point out that the flux-based methods are the most accurate, but TOAR did not include these methods because their calculation on the global scale is not yet feasible.

Considering the reviewer's comment, we have simplified the paragraph for clarity. We have deleted the information less relevant to our manuscript since conflicts with other studies on this topic.

Page 4, Line 4 "Butler et al. (2018) describes in detail" should be "Butler et al. (2018) describe in detail" Because Butler et al. indicates many people, not one.

Corrected, thank you

Page 4, line 18 I don't understand this phrase: "has been modified to enable to model capacity to be used"

Following the reviewer's comment, we re-wrote the phrase as follows: "To overcome these limits, we modified the header file gdata.h, located in ~/KPP/kpp/kpp-2.1/src. Hence, the new gdata.h file considers a large number of species and reactions associated with this new chemistry option."

Page 4, Line 19 Should this sentence be a part of the preceding paragraph?

Indeed. Thanks to the reviewer for pointing to this.

Page 5 line 31 Here and throughout you need to be consistent with regards to the term, ozone concentration. "Ozone concentration" is appropriate when using units of µg m−3, but when using units of ppb, the correct term is "ozone mixing ratio"

We corrected, thank you.

In many places in the paper the term "NOx precursors" is used. This seems to be a redundant phase. Just define NOx as a precursor gas at the beginning of the paper and then afterwards just use "NOx" as it will be clear that it is a well-known precursor.

Following the reviewer's suggestion, in the "Introduction" we define NOx precursors as NOX and the use of the "NOx" term to refer to this precursor throughout the manuscript.

Page 8, line 18 "In all receptor regions, the MDA8 O3 concentration is dominated by O3 produced by remote anthropogenic precursors" I think this is overstated because it gives the impression that far more than half of the ozone is from remote anthropogenic precursors. But the range is 30-53%. Please re-phrase this sentence.

Following the reviewer's comment, we re-phrase it as follows "In all receptor regions, local anthropogenic sources have a lower contribution to MDA8 O3 mixing ratios than the sum of ozone due to anthropogenic sources in other European source regions and long-range transport of ozone from intercontinental source regions. The contribution of intercontinental transport to the total MDA8 O3 mixing ratio in Europe is consistent with previously reported results, i.e. Fiore et al. (2009) and Karamchandani et al. (2017), while this study allows us to identify which anthropogenic sources exert a strong influence on MDA8 O3 predicted in different regions.

Page 8 line 25 This sentence seems out of place and it should appear before the discussion of the stratospheric source.

Following the reviewer's suggestion, we moved this sentence.

Page 8 line 29 "Arabic Saudi peninsula" should be Arabian Peninsula

Corrected, thank you

Page 9 line 11 This statement is not quite correct: "Another consequence of enhanced photochemical activity during the summer season is the reduction of stratospheric O3" It should say: "Another consequence of enhanced photochemical activity during the summer season is that it reduces the relative influence of stratospheric O3" But in addition to changing the relative impact, do the absolute values of stratospheric ozone also diminish in summer, relative to spring?

Indeed, the absolute ozone mixing ratio attributable to the stratosphere is reduced in our simulations in summer compared with spring. We have updated the sentence as follows: "Another consequence of enhanced photochemical activity during summer is that it reduces the influence of stratospheric O3 from a domain-wide mean MDA8 O3 mixing ratio of 4.4 ppb in the spring to 1.3 ppb in the summer (Figs. 2 and 3)."

Page 9 line 31 Is OCN in these studies defined in the same way as in your study? If not then OCN should not be used when referencing the other studies.

No, in these studies the total shipping emissions are assigned to OCN. In our study, we define three source regions associated with the shipping activities: OCN (Oceanic sources coming from boundaries and the Atlantic Ocean), MBS (the Mediterranean and the Black Sea) and BNS (Baltic and the North Sea). Following the reviewer's suggestion, we remove these references.

Page 12, line 34 I don't understand what is meant by "efficiently complementing". I think you are trying to say that they behave similarly.

Thanks for this suggestion. We change accordingly the text "Since the difference between AOT40 and SOMO35 is only a few percentage points, regardless of the receptor region, we were able to conclude that they behave similarly, according to thresholds used to define these metrics."

Page 13, line 21 Here you say that W126 is more sensitive to local emissions because it does not include a threshold. But, in addition, isn't it also more sensitive to local emissions because this metric places much more weight on the high ozone values and high ozone is likely to be produced under hot stagnant conditions when local emissions are more important?

We agree that this sentence was misleading, so we have removed it from the manuscript. We would also like to point out that on pages 13, line 32 of the original manuscript we have a paragraph that provides an explanation to the observed sensitivity of the W126 metric to higher ozone concentrations. This paragraph states "
[revised manuscript text omitted]

---

## Author Response (AR2)

Comments to the Author:

Many thanks for the revision of your manuscript. This has been improved significantly since the initial submission by addressing the comments of the reviewers and I see you have performed a deep linguistic correction, however more effort is needed in this respect. I have a number of points on this revised version that further require to be considered (pages and line refer to the track changes version) and several other minor ones. Please revise your manuscript accordingly and resubmit the re-revised track changes version together with a point-by-point reply to the comments for final consideration.

The authors want to thank the Co-Editor, Maria Kanakidou, for her valuable comments and suggestions that greatly helped us to improve the presentation of our results in the paper. The answers to specific questions are addressed below, while the modifications made in the manuscript are in blue. All Co-Editor comments are given in black, replies in blue.

Major comments:

1-      Page 2, First sentence is misleading. Nitrogen oxides do not react directly with volatile organic compounds to form ozone. To have to rephrase this sentence, for instance to 'Tropospheric ozone is formed primarily during the oxidation of volatile organic compounds in the presence of nitrogen oxides and sunlight'.

The Co-Editor's suggestion to clarify this sentence was adapted.

2- Table 1 provides the list of tagged European source regions. However in Figures 6,7, and 8 other source regions and source types are also presented. I suggest to add those in Table 1 (as 2 (?) different categories) clearly mentioning that the contribution of those comes from the boundary conditions (discussed in page 6 last paragraph).

Following the Co-Editor's comment, we revised Table 1 to account for the following categories: European source regions (already existing in the previous version of the manuscript), HTAP2 source region (we clearly state that they represent chemical boundary condition of modeled species) and global ource types.

3- How this 'tracking of contribution' is done for these extra regions and source types requires a bit more explanation than currently provided in page 6. You need also to make clear whether you double count of not between source regions and source types. Double counting should not happen and remains unclear of this is the case.

We do not double-count emissions, and we hope that this is clearer with the following modified text on page 6: "NOx emitted by several source regions and types are tagged and explicitly tracked using additional tagged reactions and tracers. Thus, we follow the contribution to the total ozone concentration from each specific emission source and type,

from both within and outside the European model domain. Table 1 summarizes the source regions and types that are used in this study. Using a division of source regions within the European model domain, 15 geographical source regions are specified in Table 1 and depicted in Figure 1. A similar division of European regions has been used by Christensen et al. (2007) and Otero et al. (2018) to address the main sources of uncertainty in regional climate simulations, as well as during the AQMEII project (Struzewska et al., 2015). Except for ALP, the source regions within the European domain are identical to receptor regions. Given the complex topography of the ALP source region, we split this region into two receptor regions: the Po Valley region and the high Alps (regions above 1500 m elevation).

To represent the impact of transported O3 from different regions outside of the domain, we used chemical boundary conditions derived from the extended CAM-Chem version 1.2 global simulations. Butler et al. (in preparation) used the tagging approach within the CAM-Chem model for several HTAP2 source regions such as: ASI (Asia), NAF (North-Africa), NAM (North-America), OCN (Oceanic sources), RBU (Russia, Belarus, Ukraine), and RST (rest of the world), as well as for several other source types: BIO (biogenic emissions), BMB (biomass burning emissions), LGT (lightning), and STR (stratospheric O3). No overlap of source regions or types is allowed.

The BIO, BMB, LGT, and STR source types are also included in the tagged chemical mechanism used in this simulation, but without including them into the division of source regions; we refer to these sources as "other global source types" from here on. Ozone due to these other global source types can originate both from long-range transport from remote source regions through the lateral model boundaries as well as from precursor emissions within the European model domain.

For each receptor region, we analyse the impact of the NOx emissions coming from different source regions and types to the total O3 concentration."

4- Table 1 and Figures 6 and 8 – please make naming of regions compatible between the Table and the Figure's titles. Also in Table 1 better explain GEN as Germany, Belgium, Netherland, Luxembourg; ALP as High Alps (West Austria…); SCA as Scandinavia (Finland…). TCA should be used in Figures instead of Turkey.

Following the Co-Editor's suggestion, we changed in Table 1 and throughout the manuscript the acronym of the CEN region to CEE to be comparable with the Figure's title. We have also updated Figure 5; now Figure 5 titles use the emission source regions and types acronym instead of the long name of these emissions sources. We have also better explained the acronyms of several source regions as suggested by the Co-Editor.

5- Page 11, lines 25-30: The change in this sentence to address the reviewer's comments

leads to a misunderstanding, leaving a wrong message to the reader. Stratospheric ozone intrusions to the troposphere vary seasonally with maxima in winter/spring. The summertime increase in photochemical production of ozone is not responsible for the changes in the stratospheric influx, which is driven by atmospheric dynamics. So the fact the stratospheric CONTRIBUTION to surface ozone levels is less during summer is due to two reasons to the decreased stratospheric influx of O3 and the increased photochemical production of ozone during that period. Therefore some rephrasing is needed at this part of the discussion.

Following the Co-Editor's comment, we rephrased the sentence as follows: "The enhanced photochemical activity during summer combined with the weakening of stratospheric-tropospheric exchange reduces the influence of stratospheric O3 from a domain-wide mean MDA8 O3 mixing ratio of 4.4 ppb in the spring to 1.3 ppb in the summer (Figs. 2 and 3)."

6- Figure 8 is impossible to read for the other source types and HTAP regions – some lumping would increase readability.

Following Co-Editor's comment, we lumped together all source types in the "Other global source types" category and all HTAP source regions in the "Aggregated HTAP source regions". We realized that the definition "source types" could confuse the reader, therefore, in the current version of the manuscript, we introduced for the first time the term "other global source types". The description of the terms can be found in the modified text in response to comment #3. The new sentence reads as follows: "The BIO, BMB, LGT, and STR source types are also included in the tagged chemical mechanism used in this simulation, but without including them into the division of source regions; we refer to these sources as "other global source types" from here on".

7- Improve Figure 6,7 and 8 captions to clearly state that contributions from other source types and HTAP regions are also shown

Following the Co-Editor recommendations, we have changed Figures 6 to 8 captions as follows:

"Figure 6. Mean modelled and observed MDA8 O3 mixing ratio filtered by a threshold of 120 ug m-3 for Po Valley, (top panel) high Alps (third from top panel) and GEN (fifth from top panel) and percent contribution to MDA8 O3 from different emissions sources and types for Po Valley (second panel), high Alps (fourth) and GEN (bottom panel) during April-September 2010 period. In each case, the contributions of tagged sources to the total O3 are shown. The tagged contributions of local and other European sources, HTAP2 source regions and other global source types to observed O3 are obtained by scaling the observed O3 by the relative contributions of these tagged sources to modelled O3. The total number of

exceedances (and non-exceedances) of the MDA8 O3 target value is indicated at the top of each column."

"Figure 7. Comparison of percent contribution of local and other European sources, HTAP2 source regions and other global source types to different O3 metrics. The metrics analysed are mean, MDA8 and 95th percentile (ppb) for the early "F" and late "S" simulation period, W126 (ppm-hours), SOMO35 and AOT40 (ppb-hours). The white dashed lines on each panel separate different categories (intercontinental transport, other global source types, and local and other European sources)"

"Figure 8. April-September 2010 time series of daytime a) hourly O3 (ppb), b) hourly AOT40 index (ppb-hours), c) hourly W126 index (ppb-hours) and d) differences between W126 and AOT40 indexes (ppb-hours) averaged over Po Valley receptor region. The color bars indicate the O3 source categories - aggregated HTAP2 regions, the other global source types and the European source regions. The white dashed lines on each panel separate different categories (intercontinental transport, other global source types, and local and other European sources)."

8- Page 19, lines 15-18: rephrase this sentence AOT40 by definition does not account at all for O3 below 20 ppb.

We want to point out that when local sources explain more than 20 ppb of the ozone mixing ratio, they contribute more to W126 than they do to AOT40. However, as this sentence is slightly confusing we revised it as follows: "We noticed that when the local sources contribute to more than 20 ppb of the O3 mixing ratios, these mixing ratios have a higher contribution to W126 than they do to AOT40 and determined that the difference was mostly due to the definition of W126 which takes into account all O3 values, not only those that are above a certain threshold."

Pages and line to check for English use and rephrasing for clarity:
9- Page 2, lines 25: are ◊ is
We corrected it, thank you.

10- Line 30: enhances
We corrected it, thank you.

11- Page3, line5 to make estimations
We corrected it, thank you.

12- Line 31-32: ugly sentence, rephrase the sentence

Following the Co-Editor's comments, we changed the text as follows: "To better understand the changes in air pollution levels, it is necessary to know the relationship between levels of an emitted species and its atmospheric concentration."

13- Line 32: information is available

We corrected it, thank you.

14- Page 5, line 16-20: rephrase

Taking into account the Co-Editor's comment, we rephrase the text as follows: "The new chemistry option considers a large number of species and reactions; therefore we exceeded hard-coded limits that the KPP chemical preprocessor, version 2.1 (Sandu and Sander, 2006) allows. To overcome these limits, we increased MAX_EQN and MAX_SPECIES in the header file gdata.h, located in: ~/KPP/kpp/kpp-2.1/src."

15- Line 18: a long term objective

We corrected it, thank you.

16- Line 27: 'briefly describe the details'? please rephrase

Following the Co-Editor's comments, we changed the text to "we discuss the details".

17- Page 5, line 22: to consider

We corrected it, thank you.

18- Page 7, lines 2-3: unclear statement –please clarify and justify that there is no double counting (see major points)

In response to the major comment 3, we explained that we do not double-count emissions. Also, as we discussed in the penultimate paragraph of Section 2.1, to avoid any numerical errors associated with the advection scheme we used a mass fixer, thus the sum of the tagged ozone is always equal to the real ozone.

19- Line 19: m-3 correct exponents

Done.

20- Page8, line 8: is (the contribution)

We corrected it, thank you.

21- Line 17 (analysis) was determined ◊ was performed

We corrected it, thank you.

22- Page 9, line 6: set-up used…

We corrected it, thank you.

23- Line16: data were well reproduced

We corrected it, thank you.

24- Line 18: temperature was …

We corrected it, thank you.

25- Page 10, line 7: different O3 sources – rephrase because O3 is secondary anyway i.e. chemically produced in the atmosphere.

We rephrased the text, taking into account the Co-Editor's comment as follows: "Since the focus of this study is on the contribution of different sources of precursors to the total tropospheric O3 concentration of a particular area, a more thorough analysis of the ability of the model to reproduce the observed meteorological variables is beyond the scope of this paper."

26- Line 13: combined contribute

Corrected

27- Line16: I am not aware of an ocean source of ozone, if you refer to emissions of NOx from shipping please rephrase.

Following the Co-Editor's suggestion, we revised the sentence as follows: "O3 from shipping NOx emissions advected through the model boundaries combined with O3 produced from shipping NOx emissions in the Atlantic Ocean mostly affects Atlantic coastal countries."

28- Line 21: other source types

Corrected

29- Line 21: this time of the year◊ better specify months

Done

30- Page 11, line3 explain boundary conditions for stratospheric O3

Following the Co-Editor's recommendation, we rephrase the text as follows: "The MOZART chemical mechanism used in this study does not explicitly treat stratospheric chemistry; thus surface stratospheric O3 could be attributed to the vertical and horizontal transport of stratospheric O3 and stratospheric tagged precursor species concentrations coming from the CAM-Chem extended model that enters the domain through lateral boundaries."

31- Line 19: noticed the spread

We corrected it, thank you.

32- Line 34: was contribution

We corrected it, thank you.

33- Page 12, line 22: peaks

We corrected it, thank you.

34- Page 13, line 1: have shown

We corrected it, thank you.

35- Line 25: type to modelled and to observed MDA8 O3 values

Done

36- Line 28: I think 'occurred' is not needed.

We agree with the Co-Editor, therefore we removed "occurred"

37- Lines 30-31: the contribution … is (or both in plural)

We corrected it, thank you.

38- Page 16, line23: natural source account

We corrected it, thank you.

39- Page 17, line9 remove 'more'

Done

40- Line 12 remove 'that'

Done

41- Line 20: Due to the way

We corrected it, thank you.

42- Line 27: linear regressions

We corrected it, thank you.

43- Line 29 more highly ? rephrase

We have replaced "more highly" with "have a higher contribution"

44- Lines 28-30: please rephrase, sentence is ugly

Following the Co-Editor's recommendation, we rephrase the text as follows: "We saw that in general, high mean O3-
[revised manuscript text omitted]